# QUALITY-SIMILAR DIVERSITY VIA POPULATION BASED REINFORCEMENT LEARNING

**Shuang Wu**[1*], **Jian Yao**[1*], **Haobo Fu**[1*], **Ye Tian**[1], **Chao Qian**[2], **Yaodong Yang**[3],
**Qiang Fu**[1], **Wei Yang**[1]
[1]Tencent AI Lab, Shenzhen, China
[2]State Key Laboratory for Novel Software Technology, Nanjing University, Nanjing, China
[3]Peking University, Beijing, China

## ABSTRACT

Diversity is a growing research topic in Reinforcement Learning (RL). Previous research on diversity has mainly focused on promoting diversity to encourage exploration and thereby improve quality (the cumulative reward), maximizing diversity subject to quality constraints, or jointly maximizing quality and diversity, known as the quality-diversity problem. In this work, we present the quality-similar diversity problem that features diversity among policies of *similar qualities*. In contrast to task-agnostic diversity, we focus on task-specific diversity defined by a set of user-specified Behavior Descriptors (BDs). A BD is a scalar function of a trajectory (e.g., the fire action rate for an Atari game), which delivers the type of diversity the user prefers. To derive the gradient of the user-specified diversity with respect to a policy, which is not trivially available, we introduce a set of BD estimators and connect it with the classical policy gradient theorem. Based on the diversity gradient, we develop a population-based RL algorithm to adaptively and efficiently optimize the population diversity at multiple quality levels throughout training. Extensive results on MuJoCo and Atari demonstrate that our algorithm significantly outperforms previous methods in terms of generating user-specified diverse policies across different quality levels (see Atari and MuJoCo videos).

## 1 INTRODUCTION

Existing research on policy diversity in deep Reinforcement Learning (RL) can be generally divided into three categories, according to the role diversity plays. The first category (Hong et al., 2018; Eysenbach et al., 2018; Conti et al., 2018; Parker-Holder et al., 2020; Kumar et al., 2020; Peng et al., 2020; Tang et al., 2020; Han & Sung, 2021; Chenghao et al., 2021; McKee et al., 2022) focuses on maximizing the final quality (the cumulative reward) of a policy, and policy diversity only serves as a means to better fulfill this goal via improving the efficiency of exploration. Therefore, the diversity measure is preferred to be task-agnostic as the knowledge of what type of task-specific diversity benefits the quality may not be accessible in most cases. The second category (Masood & Doshi-Velez, 2019; Zhang et al., 2019; Sun et al., 2020; Ghasemi et al., 2021; Zahavy et al., 2021; Zhou et al., 2022) is concerned with constrained optimization problems, where either diversity is optimized subject to quality constraints or vice-versa. Again, existing methods in this category have mainly focused on task-agnostic diversity, thereby the obtained diversity is often explained in hindsight, i.e., it is unknown what type of policy diversity to expect until the optimization is finished. The third category optimizes quality and diversity simultaneously, which is usually known as the Quality-Diversity (QD) method (Cully et al., 2015; Mouret & Clune, 2015; Pugh et al., 2016; Colas et al., 2020; Fontaine & Nikolaidis, 2021; Nilsson & Cully, 2021; Pierrot et al., 2022; Wang et al., 2021; Tjanaka et al., 2022). In contrast to task-agnostic diversity, most QD methods focus on task-specific diversity, where users are allowed to specify a set of interested Behavior Descriptors (BDs). A BD is a scalar function of a trajectory (i.e., the whole game episode) and thus does not have an analytical function form with respect to a single policy or state. Therefore, the gradient of a BD with respect to a policy is not trivially available, and this extends to the diversity measure

---

*Equal contribution. Correspondence to: Haobo Fu (haobofu@tencent.com).

defined on multiple BDs. As a result, previous QD methods (Cully et al., 2015; Mouret & Clune, 2015; Pugh et al., 2016) rely on black-box optimization techniques, such as evolutionary algorithms, to evolve a population of diverse policies. Some recent QD methods (Colas et al., 2020; Fontaine & Nikolaidis, 2021; Nilsson & Cully, 2021; Pierrot et al., 2022; Tjanaka et al., 2022) try to inject gradient information into the evolutionary optimization process.

In this work, we formulate the Quality-Similar Diversity (QSD) problem where the objective is to produce a set of diverse policies at multiple quality levels. We propose a new QD metric called the QSD score that clusters policies of similar qualities, and the diversity is evaluated at each quality level. In QSD problems, diverse policies of non-optimal qualities are also preferred, which directly meet practical needs in some real-world AI applications. For example, in the field of game AI (Zhang et al., 2021; Fu et al., 2021), it is often desirable to provide diverse accompanying AIs whose qualities are matched to a beginner, an amateur, and a master, respectively. Besides, measuring the diversity between a beginner and a master would be of little interest. The QSD problem also connects with adaptive curricula (Wang et al., 2019; Team et al., 2021; Parker-Holder et al., 2022), where the environment gradually increases curriculum levels from simple to complex. Optimizing the intermediate diversity at non-optimal quality levels helps a faster and better convergence of the agent's capabilities than training directly at a complex curriculum level.

Moreover, the ability to generate task-specific diversity is superior and supplementary to task-agnostic diversity when the user has a clear preference for the type of diversity in practice. For example, diverse hand gestures are of no interest if the user only needs gait diversity in robot locomotion tasks. Hence, in this work, we optimize an explicit diversity measure function defined on several user-specified BDs, as opposed to the non-differentiable cell coverage percentage in most QD methods. To the best of our knowledge, none of existing methods has obtained the exact gradient of a user-specified BD (defined on trajectories) with respect to a policy, nor has any derived an unbiased estimation of this gradient using state-action samples. In particular, the diversity gradient is approximated by generating samples in the policy parameter space (Colas et al., 2020; Tjanaka et al., 2022), or simply assumed in Fontaine & Nikolaidis (2021), which might not hold in many real-world situations. A set of 'state' BDs (essentially a type of intrinsic reward) are introduced in Pierrot et al. (2022), expecting that a positive correlation between state and trajectory BDs might suffice.

To fill this gap, we propose a set of *BD estimators* that predict the corresponding BD value for the current policy. Equipped with these BD estimators, we build on the policy gradient theorem (Sutton et al., 1999; Silver et al., 2014) to derive the gradient of user-specified BDs with respect to a policy for discrete or continuous actions. Based on the population-based training (PBT) (Jaderberg et al., 2017), we develop an RL diversity algorithm, named QSD-PBT, that leverages diversity gradient and adaptively adjusts diversity loss to preserve similar qualities of the population. QSD-PBT efficiently optimizes the diversity of multiple quality levels in a single run and outperforms previous methods in terms of the QSD score in both MuJoCo and Atari environments. Meanwhile, QSD-PBT demonstrates strong abilities in achieving user-specified diversity by discovering visually distinct policies across a variety of environments. To summarize, the contributions of this work are as follows:

- We formulate the Quality-Similar Diversity (QSD) problem and propose a new performance metric.
- We derive the gradient of user-specified BDs defined on trajectories with respect to a policy.
- We develop a population-based RL algorithm that efficiently optimizes the diversity of multiple quality levels in a single run.

## 2 PROBLEM DEFINITION

We focus on the episodic Markov Decision Processes (MDPs), which can be defined by a tuple $(\mathcal{S}, \mathcal{A}, \mathcal{T}, r, \gamma)$. $\mathcal{S}$ and $\mathcal{A}$ stand for the state space and action space respectively. $\mathcal{T}: \mathcal{S} \times \mathcal{A} \to \mathcal{S}$ is the environment transition function, and $r: \mathcal{S} \times \mathcal{A} \to \mathbb{R}$ is the expected reward function. A policy $\pi(s)$ maps a state $s$ to a probability distribution over $\mathcal{A}$. A trajectory $\tau$ is a state-action sequence $[s_0, a_0, s_1, a_1, ..., s_T]$, which is obtained by executing a policy from the initial step $t = 0$ to the terminal step $T$ in the environment. The objective of RL is to find a policy $\pi$ that maximizes its expected cumulative rewards (also known as the quality in this work: $J(\pi) = \mathbb{E}_{\tau \sim \pi}[R(\tau)]$, where $R(\tau) = \sum_{t=0}^{T} \gamma^t r(s_t, a_t)$ is the return of a trajectory, and $\gamma \in [0, 1]$ is the discount factor. The state value function $V^\pi(s) = \mathbb{E}[\sum_{t=i}^{T} \gamma^{t-i} r(s_t, a_t) | s_i = s]$ measures the quality following $\pi$ from state

$s$. In particular, we define $V^\pi(s_T) = 0$. The Q-function $Q^\pi(s, a) = \mathbb{E}[\sum_{t=i}^{T} \gamma^{t-i} r(s_t, a_t)|s_i = s, a_i = a]$ measures the quality following $\pi$ after taking action $a$ in state $s$.

**Behavior descriptor**. We would like to maximize a task-specific diversity measure at different quality levels in this work. Hence, users are allowed to specify a set of BDs, $b_i(\tau)$ $(1 \le i \le L)$ that reflect the type of interested policy diversity. $b_i(\tau)$ is an evaluation function of a trajectory that is finite and easy to be implemented. For instance, a BD could be the ratio between left and right movements in the trajectory of Atari MsPacMan. The trajectory BD is a general form and can be simplified to state or action BD when the user is only interested in certain states or actions of the environment, e.g., the terminal position in MsPacMan. Accordingly, the BD value of a policy $\pi_\theta$ (parameterized using $\theta$) is defined as $B_i(\pi_\theta) = \mathbb{E}_{\tau \sim \pi_\theta}[b_i(\tau)]$. We denote all the BD values of a policy $\pi_\theta$ by $\mathcal{B}(\pi_\theta) = [B_1(\pi_\theta), B_2(\pi_\theta), ..., B_L(\pi_\theta)]$.

**Diversity measure function.** Since BD values of a policy form an $L$-dimensional vector, we need a function $f$ that measures the overall diversity of the population as a scalar: $\mathrm{Div}(\Pi) := f(\mathcal{B}(\pi_{\theta_1}), \mathcal{B}(\pi_{\theta_2}), ..., \mathcal{B}(\pi_{\theta_N}))$, where $\Pi = \{\pi_{\theta_j}|1 \le j \le N\}$ denotes a set of policies. The choice of the $f$ should follow these properties: (1) $f$ should be bounded and non-negative for easy implementation; (2) Since we do not define the order of an agent in the population, $f$ should be invariant of any permutation of the policies; (3) $f$ should be differentiable such that we can derive its gradient. Two recommended measure functions are detailed in Section C.2: the mean of all pair-wise euclidean distances, and the Determinantal Point Process (DPP) (Parker-Holder et al., 2020).

**Quality-similar diversity score.** Note that the quality of a policy for a task is usually real-valued and one-dimensional, obtaining a diverse set of policies at every possible quality level would require an infinite number of policies. Hence, we approximately evaluate an algorithm's QSD performance by partitioning the obtained quality range into $M$ disjoint intervals, and only the diversity of policies within the same quality interval is evaluated by $f$. The QSD score is defined as follows:

$$\text{QSD score} := \sum_{m=1}^{M} \mathrm{Div}(\Pi_m), \tag{1}$$

where $\Pi_m$ denotes the set of policies obtained throughout training with qualities in the $m$-th interval.

## 3 POPULATION-BASED RL FOR QUALITY-SIMILAR DIVERSITY

The gradient of the user-specified BD with respect to a policy is not tractable since it is defined on a trajectory. Hence, a possible solution to our QSD problem would be adapting derivative-free methods, such as conventional QD methods (Mouret & Clune, 2015). However, derivative-free methods often scale poorly with large-scale neural networks that are necessary to handle complex inputs, such as the image feature in Atari. Note that the form in our definition of user-specified BD is similar to the quality, it would be feasible to directly derive the diversity gradient using the policy gradient theorem (Sutton et al., 1999). Given the diversity gradient, there exist two choices as how to obtain a diverse population: sequential training (Zhang et al., 2019; Zahavy et al., 2021; Zhou et al., 2022) and population-based training (Jaderberg et al., 2017; Jung et al., 2019; Parker-Holder et al., 2020). The population-based training is more appropriate in our case, because the diversity measure is defined on a population that can not be factorized into incremental diversity settings as in Zhang et al. (2019); Zahavy et al. (2021); Zhou et al. (2022). Based on the above discussion, we develop an efficient population-based RL algorithm, named QSD-PBT, for optimizing user-specified diversity BDs across different quality levels. Each component of QSD-PBT is elaborated in the following.

### 3.1 DERIVING THE DIVERSITY GRADIENT

Using the chain rule, we can write the gradient of the diversity of a population $\Pi = \{\pi_{\theta_j}|1 \le j \le N\}$ with respect to the policy $\theta_j$ (without loss of generality, we use $\theta$ hereafter) as:

$$\frac{\partial \mathrm{Div}(\Pi)}{\partial \theta} = \sum_{i=1}^{L} \frac{\partial f}{\partial B_i(\pi_\theta)} \frac{\partial B_i(\pi_\theta)}{\partial \theta}. \tag{2}$$

The partial derivative of $f$ with respect to $B_i(\pi_\theta)$ is easily obtained, as long as $f$ is an explicit diversity measure function, such as the mean pair-wise distance or the determinant in DPP. Note

that in $B_i(\pi_\theta) = \mathbb{E}_{\tau \sim \pi_\theta}[b_i(\tau)]$, $b_i(\tau)$ is a scalar function evaluating the $i$-th user-specified BD with respect to a trajectory $\tau$. Following the policy gradient theorem (Sutton et al., 1999), we have:

$$\frac{\partial B_i(\pi_\theta)}{\partial \theta} = \mathbb{E}_{\tau \sim \pi_\theta}[\sum_{t=0}^{T} b_i(\tau) \nabla_\theta \log \pi_\theta(a_t|s_t)]. \tag{3}$$

The gradient in Equation 3 is preferred to be estimated by samples $[s_t, a_t, s_{t+1}, \hat{b}_i(\tau)]$ in practice. To this end, a set of state and state-action BD estimators parameterized by $\phi_i$ are introduced:

$$V_{B_i}^{\pi_\theta}(\tau_{0:t-1}, s_t; \phi_i) := \mathbb{E}_{a_t, \tau_{t+1:T} \sim \pi_\theta}[b_i(\tau)], \tag{4}$$

$$Q_{B_i}^{\pi_\theta}(\tau_{0:t-1}, s_t, a_t; \phi_i) := \mathbb{E}_{\tau_{t+1:T} \sim \pi_\theta}[b_i(\tau)], \tag{5}$$

where $\tau_{i:j}$ denotes a segment of a trajectory starting from time step $i$ to $j$: $\tau_{i:j} = [s_i, a_i, ..., s_j, a_j]$. It is worth noting that $b_i(\tau)$ can not be factorized into a sum of quantities at each time step. For this reason, $V_{B_i}^{\pi_\theta}(\tau_{0:t-1}, s_t)$ at state $s_t$ depends on the whole historical state-action sequence $\tau_{0:t-1}$. Similar to the advantage in RL, we define the BD advantage $A_{B_i}^{\pi_\theta}(\tau_{0:t-1}, s_t, a_t) = Q_{B_i}^{\pi_\theta}(\tau_{0:t-1}, s_t, a_t) - V_{B_i}^{\pi_\theta}(\tau_{0:t-1}, s_t)$. Since $V_{B_i}^{\pi_\theta}(\tau_{0:t-1}, s_t)$ is action-independent, the gradient in Equation 3 can be estimated using a mini-batch of samples $\{(\tau_{0:t-1}^{(k)}, s_t^{(k)}, a_t^{(k)}, \hat{A}_{B_i}^{(k)})\}_{k=1}^{K}$ as:

$$\frac{\partial B_i(\pi_\theta)}{\partial \theta} \approx \frac{1}{K} \sum_{k=1}^{K} \hat{A}_{B_i}^{(k)} \nabla_\theta \log \pi_\theta(a_t^{(k)}|s_t^{(k)}), \tag{6}$$

where the sampled BD advantage $\hat{A}_{B_i}^{(k)}$ can be estimated using the conventional methods, e.g., Generalized Advantage Estimator (GAE) (Schulman et al., 2016), Direct Advantage Estimation (Pan et al., 2021). The above derives the diversity gradient for the discrete-action policy. Based on the deterministic policy gradient theorem (Silver et al., 2014), we have the following proposition for deterministic policy $\pi_\theta(s)$ in continuous action space.

**Proposition 1.** *In deterministic policy and continuous action space case, the derivative of $B_i(\pi_\theta)$ with respect to policy parameters $\theta$ is:*

$$\frac{\partial B_i(\pi_\theta)}{\partial \theta} = \sum_{t=0}^{T} \int_{s_{0:t}} p(s_0 \to s_t) \nabla_\theta \pi_\theta(s_t) \nabla_{a_t} Q_{B_i}^{\pi_\theta}(\tau_{0:t-1}, s_t, a_t)|_{a_t = \pi_\theta(s_t)} \mathrm{d}s_{0:t}, \tag{7}$$

*where $\int_{s_{0:t}}$ and $\mathrm{d}s_{0:t}$ are short for $\int_{s_0} \int_{s_1} ... \int_{s_t}$ and $\mathrm{d}s_0 \mathrm{d}s_1 ... \mathrm{d}s_t$ respectively, and $p(s_i \to s_j) = p(s_i) \prod_{k=i}^{j-1} p(s_{k+1}|s_k, \pi_\theta(s_k))$. Using a mini-batch of samples $\{(\tau_{0:t-1}^{(k)}, s_t^{(k)}, a_t^{(k)})\}_{k=1}^{K}$, we can develop an unbiased estimator of the gradient above:*

$$\frac{\partial B_i(\pi_\theta)}{\partial \theta} \approx \frac{1}{K} \sum_{k=1}^{K} \nabla_\theta \pi_\theta(s_t^{(k)}) \nabla_{a_t^{(k)}} Q_{B_i}^{\pi_\theta}(\tau_{0:t-1}^{(k)}, s_t^{(k)}, a_t^{(k)})|_{a_t^{(k)} = \pi_\theta(s_t^{(k)})}. \tag{8}$$

The proof of proposition 1 is in Appendix A.1. Based on the proposition, at each training step, we can sample a mini-batch $\{(\tau_{0:t-1}^{(k)}, s_t^{(k)}, a_t^{(k)})\}_{k=1}^{K}$ to estimate the gradient. It is worth noting that the trajectory $\tau_{0:t-1}$ may be a quite long sequence, e.g., more than 10k state-action pairs in Atari games. Hence, a feature extractor is needed to encode the trajectory. As for training BD estimators, We use the final BD value of a trajectory as the target and apply the mean squared error loss function. More implementation details are provided in Appendix C.4.

## 3.2 OVERALL TRAINING SCHEME

In the QSD problem, we focus on maximizing the diversity at different quality levels. Previous methods applied to QD problems such as the constrained optimization subject to some quality constraints are inconsistent with our objective. Another possible solution is to combine the quality loss and the diversity loss with a coefficient and adjust the coefficient using some online learning algorithms, such as bandits (Parker-Holder et al., 2020). However, it is not obvious what the online learning objective should be and how to adapt the coefficient optimally. To solve the QSD problem, we start from dealing with the sub-problem of QSD, i.e., maximizing diversity at one quality level.

**Optimizing diversity at one quality level.** Considering a population with $N$ policies, the diversity and the average quality of the population are $\mathrm{Div}(\Pi)$ and $\frac{1}{N}\sum_{j=1}^{N}J(\pi_{\theta_j})$ respectively. Each sub-problem corresponds to optimizing a combined quality loss and diversity loss with a fixed coefficient $\lambda$, saying the target weight $\lambda_\infty$. Instead of training with $\lambda_\infty$ from start to end, we let $\lambda$ start from a large initial value $\lambda_0$ and gradually decay to $\lambda_\infty$ at the end of training. Specifically, at each training step $t$, the combined loss for the population is $L_t(\Pi) = -\frac{1}{N}\sum_{j=1}^{N}J(\pi_{\theta_j}) - \lambda_t\mathrm{Div}(\Pi)$, where we require $\lambda_{t+1} < \lambda_t$ and $\lim_{t\to\infty}\lambda_t = \lambda_\infty$. This is motivated by a general observation that exploration is important in population-based RL training, and a larger $\lambda$ focuses more on the diversity and thus helps exploration. Apart from encouraging exploration throughout training, our decaying method preserves the convergence property under some assumptions, which is proved in the following.

**Proposition 2.** *Let $f(x) : \mathbb{R}^n \to \mathbb{R}, g(x) : \mathbb{R}^n \to \mathbb{R}$ be Lipschitz smooth, convex, bounded functions with derivative bounded, and a positive decreasing series $\{\lambda_t\}_{t=0}^{\infty}$ converges to $\lambda$. Denote $h_t(x) = f(x) + \lambda_t g(x)$. Then using the gradient descent algorithm and choosing a proper step size, the algorithm will converge to the global minimum of $f(x) + \lambda g(x)$.*

The proof is in Appendix A.2. We can see that the decaying method converges in convex and Lipschitz smooth cases. In non-convex cases, we validate it experimentally in Section 4 and Appendix B.1.

**Preserving the quality similarity.** Constraining only the average quality of the population at certain levels does not guarantee similar qualities in the population. To encourage quality similarity among agents in a population, we distribute $\lambda_t$ to each agent in the population, denoted as $\lambda_t^{(j)}$ for the $j$-th agent, and adapt each $\lambda_t^{(j)}$ during training. The loss function for the $j$-th agent (policy) is $L_t^{(j)}(\pi_{\theta_j}) = -J(\pi_{\theta_j}) - \lambda_t^{(j)}\mathrm{Div}(\Pi)$. Specifically, policies with better qualities should focus more on diversity optimization, and vice-versa. Following the discussion above, the scheme of $\lambda_t^{(j)}$ throughout the training process is designed as $\lambda_t^{(j)} = \lambda_\infty + (\lambda_0 - \lambda_\infty)\exp(-\frac{t}{t_0} \cdot \frac{\max_j R_t^{(j)}}{R_t^{(j)}})$, where $\lambda_0$ is an initial coefficient. $t_0$ indicates the preferred decay step, and $t$ is the current training step. $R_t^{(j)}$ is the evaluation of the current quality of the $j$-th policy in the population. We assume that the quality of any policy is non-negative and upper bounded. As a result, we have $\lim_{t\to\infty}\lambda_t^{(j)} = \lambda_\infty$.

**Optimizing diversity across multiple quality levels.** Once a target $\lambda_\infty$ is designated, the objective function $L_t(\Pi)$ corresponds to maximizing diversity at a certain quality level. Since our goal in QSD is to obtain policy diversity at multiple quality levels, a straightforward way is to optimize the objective function $L_t(\Pi)$ multiple times with a set of target weights $\{\lambda_{\infty,h}\}_{h=1}^{H}$. Alternatively, we could obtain policy diversity at multiple quality levels in a single run. Given two target weights $\lambda_{\infty,1} > \lambda_{\infty,2}$, when we have finished the optimization with $\lambda_{\infty,1}$, we can continue the optimization from $\lambda_{\infty,1}$ to solve the quality level targeted by $\lambda_{\infty,2}$, rather than re-initializing the model and training from the initial $\lambda_0$. This can be extended to multiple target weights, if $\lambda$ is decayed slowly enough from a large value $\lambda_0$ to 0. In practice, we find it much more efficient than independent training with multiple target values $\{\lambda_{\infty,h}\}_{h=1}^{H}$. As a result, we apply a single training with $\lambda$ decaying from $\lambda_0$ to 0 to integrally solve the QSD problem, which gives the overall training loss for the $j$-th policy as :

$$L_t^{(j)}(\pi_{\theta_j}) = -J(\pi_{\theta_j}) - \lambda_t^{(j)}\mathrm{Div}(\Pi), \tag{9}$$

with $\lambda_t^{(j)} = \lambda_0 \cdot \exp(-\frac{t}{t_0} \cdot \frac{\max_j R_t^{(j)}}{R_t^{(j)}})$. We employ PPO (Schulman et al., 2017) as the backbone for discrete actions. As for scenarios with continuous action space, the TD3 (Fujimoto et al., 2018) algorithm is applied as the backbone. We term our diversity optimization along with population-based RL backbones as QSD-PBT. The pseudocode is given in Appendix C.5, and the code is open-sourced.

## 4 EXPERIMENTS

The effectiveness of QSD-PBT is validated on both MuJoCo (Brockman et al., 2016) continuous control tasks and Atari games (Bellemare et al., 2013) with discrete action spaces. For the diversity measure, we use the mean of pair-wise Euclidean distance between the BD vectors of two policies. For the specification of BDs, we follow common practices in MuJoCo tasks by incorporating the built-in joint torques as BDs, similar to Parker-Holder et al. (2020). For Atari games, we estimate the advantage by the widely-adopted GAE method and follow the PPO Schulman et al. (2017)

implementation that normalizes the advantage to make the training more robust. We design general-proposed BDs from the perspective of human players for the five Atari games. It is worth noting that the design of all the BDs does not unfairly favor any particular algorithm. Hence, the comparison among different methods is fair and unbiased (also illustrated in Appendix B.3). We compare QSD-PBT with independent PPO (Schulman et al., 2017) (driven only by the quality), two QD-style methods: QD-PG (Pierrot et al., 2022) (diversity gradient by 'state' BDs) and EDO-CS (Wang et al., 2021) (diversity gradient by evolution strategies), and two population-based RL algorithms: PBT (Jaderberg et al., 2017) (driven only by the quality) and DvD (Parker-Holder et al., 2020) (driven by both the quality and task-agnostic diversity).

For each environment, the range of quality is firstly estimated by training a state-of-the-art quality-driven method (TD3 for MuJoCo tasks and PPO for Atari games) and then partitioned into $M = 10$ disjoint intervals of equal scope. The highest quality $R_{max}$ achieved by TD3 or PPO is considered as the 'optimal' quality and shared by all the comparing methods. To calculate the QSD score for each method, we save a number of candidate policies within each quality interval during training. The time overhead for each method is estimated by the number of training steps till the average quality of the population reaches a near-optimal quality ($0.9R_{max}$). Error bars plotted in the figures or standard deviations presented in the tables are obtained using 5 independent runs. The population size $N$ is 8 for MuJoCo experiments and 10 for Atari experiments. Other implementation details are presented in Appendix C. Additional results that demonstrate the versatility of QSD-PBT for different user-specified BDs and diversity measure functions are included in Appendix B.3 and B.1.2.

## 4.1 MuJoCo Tasks

Three tasks from the MuJoCo environment are selected: Humanoid, Hopper, and Walker2d. We define two types of BDs: (1) **Scoring speed** is the final score divided by the number of time steps used, indicating how fast the agent scores. (2) **Built-in joint torques** are the average torques applied to the hinge joints over a trajectory, which have a dimension of 17, 3, and 6 respectively.

Table 1: The QSD scores on MuJoCo tasks. #step denotes the time overhead of each method.

| MuJoCo Tasks | | QD-PG | EDO-CS | PBT | DvD-TD3 | QSD-PBT (Ours) |
|---|---|---|---|---|---|---|
| Humanoid $R_{max}$: 5.5k | score > 0% | $1.78 \pm 0.12$ | $1.56 \pm 0.13$ | $1.20 \pm 0.20$ | $3.02 \pm 0.10$ | $\mathbf{3.45 \pm 0.07}$ |
| | score > 60% | $0.40 \pm 0.02$ | $0.62 \pm 0.03$ | $0.30 \pm 0.15$ | $1.13 \pm 0.02$ | $\mathbf{1.39 \pm 0.03}$ |
| | #step (k) | $760 \pm 160$ | $782 \pm 176$ | $120 \pm 1$ | $107 \pm 7$ | $\mathbf{90 \pm 15}$ |
| Hopper $R_{max}$: 3k | score > 0% | $0.59 \pm 0.01$ | $0.72 \pm 0.09$ | $0.63 \pm 0.03$ | $0.80 \pm 0.02$ | $\mathbf{1.26 \pm 0.08}$ |
| | score > 60% | $0.08 \pm 0.01$ | $0.18 \pm 0.02$ | $0.17 \pm 0.01$ | $0.24 \pm 0.01$ | $\mathbf{0.26 \pm 0.01}$ |
| | #step (k) | $222 \pm 31$ | $200 \pm 27$ | $\mathbf{111 \pm 7}$ | $208 \pm 27$ | $150 \pm 19$ |
| Walker2d $R_{max}$: 5k | score > 0% | $0.91 \pm 0.04$ | $0.86 \pm 0.10$ | $0.78 \pm 0.03$ | $1.68 \pm 0.16$ | $\mathbf{2.28 \pm 0.05}$ |
| | score > 60% | $0.07 \pm 0.01$ | $0.13 \pm 0.06$ | $0.19 \pm 0.01$ | $0.47 \pm 0.04$ | $\mathbf{0.79 \pm 0.02}$ |
| | #step (k) | $350 \pm 51$ | $471 \pm 102$ | $\mathbf{155 \pm 2}$ | $191 \pm 2$ | $186 \pm 14$ |

Table 1 shows each method's QSD score, where we also report the QSD score calculated using only intervals with high quality (above 60% of $R_{max}$). PBT gets the lowest 3 out of 6 QSD scores, indicating that methods considering only quality result in considerable degradation of diversity. QD-PG gets trapped in terms of quality, which results in a low QSD score. EDO-CS performs slightly better than QD-PG, which is consistent with the experimental results in Wang et al. (2021). QD-PG and EDO-CS are much more time-consuming than other RL-style methods. The performance of DvD-TD3 (Parker-Holder et al., 2020) comes as the second best, and QSD-PBT consistently achieves the highest QSD scores. Similar conclusions can be arrived at in Figure 1, where we plot the quality-diversity curves throughout training. One important reason for the superiority of QSD-PBT is the unbiased diversity gradient with respect to the user-specified diversity and the estimation using samples in the state-action space, while other methods are either biased (QD-PG using 'state' BDs and DvD-TD3 using task-agnostic BDs) or estimated using samples in the policy parameter space (EDO-CS). Another reason is that the diversity is better exploited within each quality interval due to the adaptive diversity loss, which is further studied in Appendix B.1. Additional illustration of the diversity at different quality levels is in Figure 8.

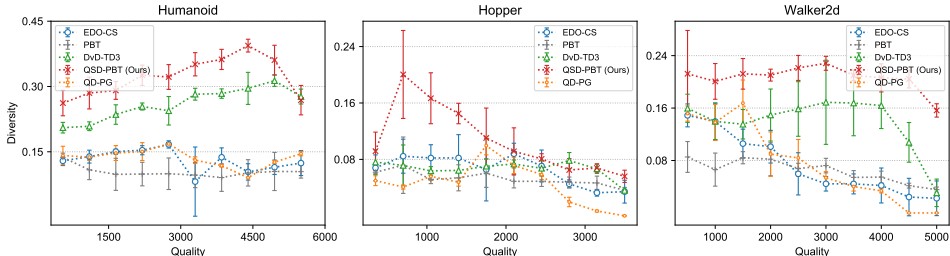

Figure 1: The quality similar diversity across 10 quality intervals on MuJoCo tasks.

## 4.2 ATARI GAMES

We select five Atari games. MsPacMan and RiverRaid are PvE (i.e., single-agent) games where players obtain scores by achieving pre-designed goals. FishingDerby, IceHockey, and Boxing are PvP (i.e., multiagent) games where the final score is the score difference between the player and a built-in AI competitor. Five general-proposed BDs are constructed from the perspective of human players: (1) **Game time** is the number of time steps in an episode. (2) **Fire rate** is the frequency of *fire* action in an episode. (3) **Action continuity** measures the continuity of an agent's actions by counting the number of action changes, which is actually an AI version of APM (actions per minute, widely applied to human players). For example, this BD for the action sequence {*noop-noop-left-left-noop*} is 2. (4) Two orientation BDs (**left_right** and **up_down**) measure the preference of moving directions.

Table 2: The QSD scores on Atari games. #step denotes the time overhead of each method.

| Atari Games | | | PPO | PBT | DvD-PPO | QSD-PBT (Ours) |
|---|---|---|---|---|---|---|
| PvE | MsPacMan $R_{max}$: 15k | score > 0% | $1.28 \pm 0.13$ | $1.14 \pm 0.09$ | $1.40 \pm 0.05$ | $\mathbf{2.47 \pm 0.06}$ |
| | | score > 60% | $0.53 \pm 0.06$ | $0.42 \pm 0.07$ | $0.56 \pm 0.04$ | $\mathbf{1.02 \pm 0.04}$ |
| | | #step (k) | $456 \pm 86$ | $\mathbf{269 \pm 52}$ | $415 \pm 52$ | $902 \pm 108$ |
| | RiverRaid $R_{max}$: 15k | score > 0% | $0.92 \pm 0.12$ | $0.84 \pm 0.08$ | $1.32 \pm 0.12$ | $\mathbf{2.66 \pm 0.13}$ |
| | | score > 60% | $0.43 \pm 0.08$ | $0.38 \pm 0.05$ | $0.60 \pm 0.05$ | $\mathbf{1.32 \pm 0.04}$ |
| | | #step (k) | $204 \pm 31$ | $\mathbf{154 \pm 36}$ | $352 \pm 144$ | $734 \pm 183$ |
| PvP | FishingDerby $R_{max}$: 48 | score > 0% | $1.60 \pm 0.17$ | $1.55 \pm 0.13$ | $2.32 \pm 0.08$ | $\mathbf{3.64 \pm 0.12}$ |
| | | score > 60% | $0.78 \pm 0.09$ | $0.76 \pm 0.06$ | $1.01 \pm 0.06$ | $\mathbf{1.68 \pm 0.05}$ |
| | | #step (k) | $262 \pm 138$ | $\mathbf{109 \pm 25}$ | $168 \pm 87$ | $242 \pm 25$ |
| | IceHockey $R_{max}$: 40 | score > 0% | $1.68 \pm 0.16$ | $1.04 \pm 0.14$ | $2.74 \pm 0.30$ | $\mathbf{4.81 \pm 0.27}$ |
| | | score > 60% | $0.81 \pm 0.05$ | $0.43 \pm 0.07$ | $1.13 \pm 0.12$ | $\mathbf{1.92 \pm 0.08}$ |
| | | #step (k) | $551 \pm 100$ | $520 \pm 152$ | $\mathbf{443 \pm 42}$ | $404 \pm 61$ |
| | Boxing $R_{max}$: 100 | score > 0% | $0.56 \pm 0.08$ | $0.69 \pm 0.07$ | $0.92 \pm 0.28$ | $\mathbf{1.28 \pm 0.17}$ |
| | | score > 60% | $0.24 \pm 0.07$ | $0.30 \pm 0.05$ | $0.40 \pm 0.13$ | $\mathbf{0.59 \pm 0.09}$ |
| | | #step (k) | $24 \pm 4$ | $\mathbf{23 \pm 1}$ | $29 \pm 7$ | $27 \pm 3$ |

In Atari games we find that the quality of QD-PG and EDO-CS algorithms reach much lower levels than other PPO-based methods, therefore we have not included them for comparison and introduced a PPO baseline. The results are summarized in Table 2 and Figure 2, most of which are consistent with that in the MuJoCo experiment. The exploitation strategy in PBT accelerates the training of quality at a considerable cost of diversity. DvD-PPO performs relatively well, even though it optimizes the task-agnostic diversity defined on action probabilities. Yet, the BDs for Atari games are more macroscopical than those for MuJoCo tasks, so the improvement of DvD-PPO over PBT is smaller in Atari games than that in MuJoCo tasks. Due to our diversity gradient and the adaptive diversity loss, QSD-PBT algorithm consistently achieves the highest QSD scores, though with more time overhead in MsPacMan and RiverRaid due to a relatively large diversity loss coefficient at the beginning.

We further demonstrate the ability of QSD-PBT in generating user-specified diversity, using the two orientation diversity BDs in MsPacMan, as shown in Figure 3 and in the video. In this game, there are shortcuts connecting the left and right ends of the maze, which allows an agent to return to the middle of the maze via consistent (e.g., always left) moving directions. The orientation preference is a macroscopic BD that is significantly different from any task-agnostic diversity investigated in

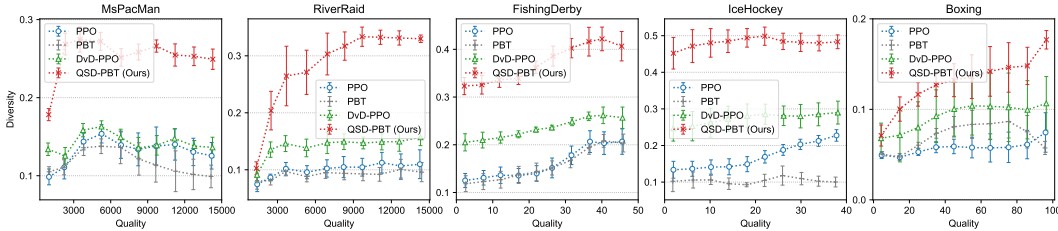

Figure 2: The quality similar diversity across 10 quality intervals on Atari games.

previous literature. As shown in Figure 3, QSD-PBT achieves diverse (in a user-specified way) and visually distinct policies with good quality while other comparing methods perform unsatisfactorily (including DvD-PPO which optimizes a task-agnostic diversity).

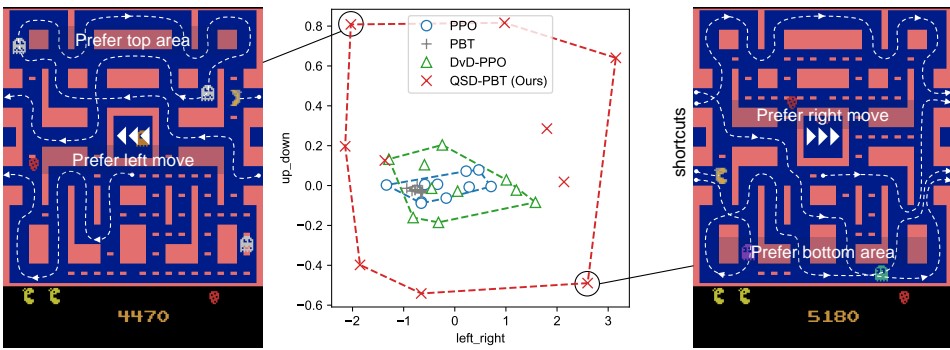

Figure 3: Visualization of orientation diversity in MsPacMan.

## 5 RELATED WORK

**Diversity as a means to improve quality.** A diversity-driven approach for exploration (Hong et al., 2018) was proposed by adding a distance regularization between the current policy and a previous policy. Unsupervised learning of diverse policies was studied in Eysenbach et al. (2018) to serve as an effective pretraining mechanism for downstream RL tasks. Novelty search was hybridized with the OpenAI ES to improve exploration in sparse or deceptive deep RL tasks (Conti et al., 2018). A population determinant diversity measure (Parker-Holder et al., 2020) was proposed to improve exploration. Diverse opponent policies have a large influence on the online performance of opponent modelling Fu et al. (2022).

Diverse behaviors were learned in order to effectively generalize to varying environments that are different from training (Kumar et al., 2020). A diversity-regularized collaborative exploration strategy was proposed in Peng et al. (2020). Reward randomization (Tang et al., 2020) was employed to discover diverse policies in multi-agent games with the aim of improving the final policy quality. Trajectory diversity was maximized for better zero-shot coordination in a collaborative multi-agent environment (Lupu et al., 2021). Diversity was studied in multi-agent open-ended learning (Liu et al., 2021; Perez-Nieves et al., 2021) to improve the final exploitability (another definition of quality).

**Maximizing diversity subject to high quality constraints or vice-versa.** A maximum mean discrepancy regularizer was proposed to produce a set of near-optimal policies having different distributions of trajectories (Masood & Doshi-Velez, 2019). A two-objective update technique (Zhang et al., 2019) was developed for sequentially obtaining a set of novel policies, each of which solves a given task in the meantime executing distinct action sequences. A method based on the Frank-Wolfe algorithm (Frank & Wolfe, 1956) was introduced to compute a set of diverse and near-optimal policies (Ghasemi et al., 2021). A set of diverse policies in the space of successor features (Barreto et al., 2017) were sequentially obtained by solving a line of constrained MDPs (Zahavy et al., 2021), where an intrinsic diversity reward was maximized subject to a quality constraint. A reward-switching

technique was recently proposed (Zhou et al., 2022) to discover a diverse set of high-quality policies by sequentially solving a novelty-constrained optimization problem for the current policy.

**Maximizing both quality and diversity via QD-style methods.** QD algorithms (Pugh et al., 2016; Cully & Demiris, 2017) are a type of evolutionary algorithms, where the goal is to evolve a set of diverse and high-quality solutions in a single run. A representative QD algorithm is MAP-Elites (Cully et al., 2015; Mouret & Clune, 2015). In order to scale to large policy neural networks, several recent attempts have been made to combine QD with policy gradient (Cideron et al., 2020; Nilsson & Cully, 2021; Pierrot et al., 2022) or ES (Colas et al., 2020; Wang et al., 2021). Evolutionary multi-objective deep reinforcement learning was employed (Shen et al., 2020) to generate behavior-diverse game AIs. A kernel-based method with Stein variational gradient descent was proposed (Gangwani et al., 2020) for training a set of QD policies. Assuming that both the quality and the BD are fully differentiable (which is generally not true in the RL settings), a new MAP-Elites algorithm was developed in Fontaine & Nikolaidis (2021). A subsequent work (Tjanaka et al., 2022) extended differentiable QD (Fontaine & Nikolaidis, 2021) to RL settings, where the quality and BD gradients were estimated using TD3 and ES respectively.

## 6 DISCUSSION AND CONCLUSION

In this work, motivated by the practical need in generating task-specific diverse policies of similar qualities, we formulate the QSD problem and a new performance metric called the QSD score. For the first time, we derive the gradient of the user-defined diversity measure with respect to a policy and approximate the gradient using samples in the state action space (as opposed to the policy parameter space). Based on our diversity gradient, an efficient population-based RL algorithm (i.e., QSD-PBT) is then developed and has demonstrated strong performance, in terms of maximizing user-specified diversity across different quality levels, on both MuJoCo and Atari.

**BD definition.** We are aware in some real-world situations it might be non-trivial to express the user-intended diversity via a set of BDs but relatively easy via a similarity function $\text{Sim}(\tau_i, \tau_j)$ between two trajectories. $\text{Sim}(\tau_i, \tau_j)$ is more general than BDs, since we can derive $\text{Sim}(\tau_i, \tau_j)$ using BDs but the opposite is not true. We look forward to extending QSD-PBT to such and more general situations.

**BD estimator.** In order to estimate our diversity gradient, we need to train the state BD (Equation 4) or state-action BD estimators (Equation 5). Note that both BD estimators depend on previous state-action sequence $\tau_{0:t-1}$, which might cause problems in very long trajectories. It is currently handled by applying LSTM models (in MuJoCo) or sufficient statistics (in Atari). More advanced techniques, such as attention (Vaswani et al., 2017), may be needed for more accurate estimation.

**Population-based training.** We use a separate neural network for each agent in the population, which increases the overall training time and memory overhead linearly with the population size $N$. A more efficient approach may be sharing most of the feature extraction part, e.g., the 3-layer convolution model in DQN, for all agents and building separate policy and value heads for each agent in the population. This implementation is connected with the ensemble and multi-task learning (An et al., 2021; Flet-Berliac & Preux, 2019). From experience in previous research, parameter sharing accelerates training convergence yet in our case degrades the diversity of PPO and PBT, since they are optimized without explicit diversity objectives. For fair comparisons, parameter sharing is not implemented in this paper but is recommended in practice.

**Quality and diversity optimization.** In this paper, we break the QSD problem into a set of sub-problems (each corresponding to maximizing diversity at a certain quality level) and solve them integrally in a single run, and the resulting algorithm is termed QSD-PBT. In spite of achieving better empirical performance than other baselines, whether QSD-PBT is the most efficient method in optimizing the QSD score defined in Equation 1 is unclear. One possible solution is combining the strength of our diversity gradient (Equation 6 and 8) with existing QD-style methods. Another is regarding the QSD problem as a Pareto front optimization problem and applying continuous exploration (Ma et al., 2020).

ACKNOWLEDGEMENT

We thank Ke Xue (from Nanjing University) and Peng Yang (from Southern University of Science and Technology) for their helpful discussions. Peng Yang and Chao Qian are sponsored by the CCF-Tencent Open Research Fund (CCF-Tencent RAGR20220110). Yaodong Yang is sponsored by the CCF-Tencent Open Research Fund (CCF-Tencent RAGR20220109). We are grateful to the anonymous reviewers for their insightful feedback.

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

# A    THEORY ANALYSIS

## A.1    PROOF OF PROPOSITION 1

*Proof.* We prove the correctness of Equation 7 and that Estimation 8 is an unbiased estimation of the gradient in Equation 7 with a constant scale. We denote a segmentation of a trajectory starting from time step $i$ to $j$ as $\tau_{i:j} := [s_i, a_i, ..., s_j, a_j]$, with specially $\tau_{-1} := \varnothing$ and $\tau_0 := [s_0, a_0]$. Recall the definitions of state BD estimator and state-action BD estimator:

$$V^{\pi_\theta}_{B_i}(\tau_{0:t-1}, s_t) := \mathbb{E}_{a_t, \tau_{t+1:T} \sim \pi_\theta}[b_i(\tau)], \tag{10}$$

$$Q^{\pi_\theta}_{B_i}(\tau_{0:t-1}, s_t, a_t) := \mathbb{E}_{\tau_{t+1:T} \sim \pi_\theta}[b_i(\tau)]. \tag{11}$$

We consider a determinate policy for continuous actions, and the output of the policy $\pi_\theta(s_t)$ is a scalar, i.e., $a_t = \pi_\theta(s_t)$. As a result, the following two equations hold between the state BD estimator and the state-action BD estimator:

$$Q^{\pi_\theta}_{B_i}(\tau_{0:t-1}, s_t, a_t) = \int_{s_{t+1}} p(s_{t+1}|s_t, \pi_\theta(s_t)) V^{\pi_\theta}_{B_i}(\tau_{0:t}, s_{t+1}) \mathrm{d}s_{t+1}, \tag{12}$$

$$V^{\pi_\theta}_{B_i}(\tau_{0:t-1}, s_t) = Q^{\pi_\theta}_{B_i}(\tau_{0:t-1}, s_t, \pi_\theta(s_t)), \tag{13}$$

where $p(s_{t+1}|s_t, \pi_\theta(s_t))$ is the transition probability from $s_t$ to $s_{t+1}$ after taking the action $\pi_\theta(s_t)$. We use $p^\pi(s_i \to s_j)$ to represent the probability $p(s_i) \prod_{k=i}^{j-1} p(s_{k+1}|s_k, \pi_\theta(s_k))$. We omit the index $\pi$ for clarity in the following. Note that the BD of a policy can be written as:

$$B_i(\pi_\theta) = \int_{s_0} p(s_0) Q^{\pi_\theta}_{B_i}(\tau_{-1}, s_0, \pi_\theta(s_0)) \mathrm{d}s_0, \tag{14}$$

and the gradient is:

$$\frac{\partial B_i(\pi_\theta)}{\partial \theta} = \int_{s_0} p(s_0) \nabla_\theta Q^{\pi_\theta}_{B_i}(\tau_{-1}, s_0, \pi_\theta(s_0)) \mathrm{d}s_0. \tag{15}$$

Further, we have:

$$\nabla_\theta Q^{\pi_\theta}_{B_i}(\tau_{-1}, s_0, \pi_\theta(s_0)) = \nabla_\theta \int_{s_1} p(s_1|s_0, \pi_\theta(s_0)) V^{\pi_\theta}_{B_i}(\tau_0, s_1) \mathrm{d}s_1$$

$$= \int_{s_1} \nabla_\theta \pi_\theta(s_0) \nabla_{a_0} p(s_1|s_0, a_0)|_{a_0 = \pi_\theta(s_0)} V^{\pi_\theta}_{B_i}(\tau_0, s_1) \mathrm{d}s_1 + \int_{s_1} p(s_1|s_0, \pi_\theta(s_0)) \nabla_\theta V^{\pi_\theta}_{B_i}(\tau_0, s_1) \mathrm{d}s_1$$

$$= \nabla_\theta \pi_\theta(s_0) \int_{s_1} \nabla_{a_0} p(s_1|s_0, a_0)|_{a_0 = \pi_\theta(s_0)} V^{\pi_\theta}_{B_i}(\tau_0, s_1) \mathrm{d}s_1 + \int_{s_1} p(s_1|s_0, \pi_\theta(s_0)) \nabla_\theta V^{\pi_\theta}_{B_i}(\tau_0, s_1) \mathrm{d}s_1$$

$$= \nabla_\theta \pi_\theta(s_0) \nabla_{a_0} Q^{\pi_\theta}_{B_i}(\tau_{-1}, s_0, a_0)|_{a_0 = \pi_\theta(s_0)} + \int_{s_1} p(s_1|s_0, \pi_\theta(s_0)) \nabla_\theta Q^{\pi_\theta}_{B_i}(\tau_0, s_1, \pi_\theta(s_1)) \mathrm{d}s_1. \tag{16}$$

Equation 16 shows the relation between the gradient of the current state-action BD estimator and that of the next state-action BD estimator. Generalizing this result, we have

$$\nabla_\theta Q^{\pi_\theta}_{B_i}(\tau_{0:t-1}, s_t, \pi_\theta(s_t)) = \nabla_\theta \pi_\theta(s_t) \nabla_{a_t} Q^{\pi_\theta}_{B_i}(\tau_{0:t-1}, s_t, a_t)|_{a_t = \pi_\theta(s_t)}$$

$$+ \int_{s_{t+1}} p(s_{t+1}|s_t, \pi_\theta(s_t)) \nabla_\theta Q^{\pi_\theta}_{B_i}(\tau_{0:t}, s_{t+1}, \pi_\theta(s_{t+1})) \mathrm{d}s_{t+1}. \tag{17}$$

We can apply Equation 17 to Equation 15 recursively:

$$
\begin{aligned}
\frac{\partial B_i(\pi_\theta)}{\partial \theta} &= \int_{s_0} p(s_0)\nabla_\theta \pi_\theta(s_0)\nabla_{a_0}Q_{B_i}^{\pi_\theta}(\tau_{-1}, s_0, a_0)|_{a_0=\pi_\theta(s_0)}\mathrm{d}s_0 \\
&\quad + \int_{s_1}\int_{s_0} p(s_0)p(s_1|s_0,\pi_\theta(s_0))\nabla_\theta Q_{B_i}^{\pi_\theta}(\tau_0, s_1, \pi_\theta(s_1))\mathrm{d}s_0\mathrm{d}s_1 \\
&= \int_{s_0} p(s_0)\nabla_\theta \pi_\theta(s_0)\nabla_{a_0}Q_{B_i}^{\pi_\theta}(\tau_{-1}, s_0, a_0)|_{a_0=\pi_\theta(s_0)}\mathrm{d}s_0 \\
&\quad + \int_{s_{0:1}} p(s_0\to s_1)\nabla_\theta Q_{B_i}^{\pi_\theta}(\tau_0, s_1, \pi_\theta(s_1))\mathrm{d}s_{0:1} \\
&= \int_{s_0} p(s_0)\nabla_\theta \pi_\theta(s_0)\nabla_{a_0}Q_{B_i}^{\pi_\theta}(\tau_{-1}, s_0, a_0)|_{a_0=\pi_\theta(s_0)}\mathrm{d}s_0 \\
&\quad + \int_{s_{0:1}} p(s_0\to s_1)\nabla_\theta \pi_\theta(s_1)\nabla_{a_1}Q_{B_i}^{\pi_\theta}(\tau_0, s_1, a_1)|_{a_1=\pi_\theta(s_1)}\mathrm{d}s_{0:1} \\
&\quad + \int_{s_{0:2}} p(s_0\to s_2)\nabla_\theta Q_{B_i}^{\pi_\theta}(\tau_{0:1}, s_2, \pi_\theta(s_2))\mathrm{d}s_{0:2} \\
&= ... \\
&= \sum_{t=0}^{T} \int_{s_{0:t}} p(s_0\to s_t)\nabla_\theta \pi_\theta(s_t)\nabla_{a_t}Q_{B_i}^{\pi_\theta}(\tau_{0:t-1}, s_t, a_t)|_{a_t=\pi_\theta(s_t)}\mathrm{d}s_{0:t}, \quad (18)
\end{aligned}
$$

where $\int_{s_{0:t}}$ is short for $\int_{s_0}\int_{s_1}...\int_{s_t}$ and $\mathrm{d}s_{0:t}$ is short for $\mathrm{d}s_0\mathrm{d}s_1...\mathrm{d}s_t$. The first equality is according to Equation 16, and the successive equality is according to Equation 17. $\qquad\square$

*Proof.* We further prove that with a constant scale, Equation 8 is an unbiased estimation of Equation 7. Note that in practice we obtain sampled trajectories by executing policy $\pi(\theta)$, and the samples in the trajectories are added into a replay buffer. As a result, a sample $(s_t, a_t, \tau_{0:t-1})$ is added into the replay buffer with a probability $p(s_0)\prod_{k=0}^{t-1}p(s_{k+1}|s_k,\pi_\theta(s_k))$, i.e., $p(s_0\to s_t)$. In other words, the probability that the sample $(s_t, a_t, \tau_{0:t-1})$ is in the current mini-batch is proportional to $p(s_0\to s_t)$, which we denote by $\frac{p(s_0\to s_t)}{C}$ ($C$ is a constant). The expectation of the sampled gradient is:

$$
\begin{aligned}
&\mathbb{E}\left[\nabla_\theta \pi_\theta(s_t)\nabla_{a_t}Q_{B_i}^{\pi_\theta}(\tau_{0:t-1}, s_t, a_t)|_{a_t=\pi_\theta(s_t)}\right] \\
&= \sum_{t=0}^{T}\int_{s_{0:t}}\nabla_\theta \pi_\theta(s_t)\nabla_{a_t}Q_{B_i}^{\pi_\theta}(\tau_{0:t-1}, s_t, a_t)|_{a_t=\pi_\theta(s_t)}\frac{p(s_0\to s_t)}{C}\mathrm{d}s_{0:t} \\
&= \frac{1}{C}\frac{\partial B_i(\pi_\theta)}{\partial \theta}, \quad (19)
\end{aligned}
$$

where the last equality is according to Equation 18. $\qquad\square$

### A.2 PROOF OF PROPOSITION 2

*Proof.* Denote $h(x) = f(x) + \lambda g(x)$. By assumption, we have $h(x)$ is Lipschitz smooth, i.e., there exists $L > 0$, s.t.

$$||\nabla h(x) - \nabla h(y)|| \leq L||x - y||, \forall x, y \in \mathbb{R}^n \quad (20)$$

So,

$$
\begin{aligned}
|h(y) - h(x) - \nabla h(x)^T(y-x)| &= |\int_0^1 \nabla h(x + t(y-x))^T(y-x)\mathrm{d}t - \nabla h(x)^T(y-x)| \\
&\leq \int_0^1 ||\nabla h(x+t(y-x))^T - \nabla h(x)^T|| \cdot ||y-x||\mathrm{d}t \\
&\leq \int_0^1 tL||y-x||^2\mathrm{d}t \\
&= \frac{L}{2}||y-x||^2 \quad (21)
\end{aligned}
$$

i.e.,

$$h(y) - h(x) \leq \nabla h(x)^T (y - x) + \frac{L}{2} ||x - y||^2. \tag{22}$$

Since $g(x)$ is bounded, we can assume $g(x) > 0$. Otherwise, let $C = \min_x g(x)$, we rewrite $h_t(x) = f(x) + \lambda_t(g(x) - C) + \lambda_t C$ and perform the following analysis on $h_t(x)_{new} = f(x) + \lambda_t(g(x) - C)$ (note that both $h_t(x)_{new}$ and $h_t(x)$ have the same derivatives and minimums). Hence, for each fixed $x$, the series $\{h_t(x)\}_{t=0}^{\infty}$ is decreasing to $h(x)$.

Consider an optimizing algorithm which starts at $x_0$, and, at each time step $t$, updates $x_t$ using the gradient descent $x_{t+1} = x_t - \eta \nabla h_t(x_t)$. By Equation 22, we have,

$$h_t(x_{t+1}) - h_t(x_t) \leq \nabla h_t(x_t)^T (x_{t+1} - x_t) + \frac{L}{2} ||x_{t+1} - x_t||^2$$

$$\leq -\eta ||\nabla h_t(x_t)||^2 + \frac{L\eta^2}{2} ||\nabla h_t(x_t)||^2 \tag{23}$$

Choosing $\eta = \frac{1}{L}$, we have, $h_t(x_{t+1}) \leq h_t(x_t)$, and since $h_{t+1}(x_{t+1}) \leq h_t(x_{t+1})$, we obtain $h_{t+1}(x_{t+1}) \leq h_t(x_t)$. Hence $\lim_{t\to\infty} h_t(x_t)$ exists ($h(x)$ is bounded). It is easy to prove that $\{h_t(x)\}_{t=0}^{\infty}$ converge to $h(x)$ uniformly, hence, $\lim_{t\to\infty} h_t(x_t) = \lim_{t\to\infty} h(x_t)$. We denote $M = \lim_{t\to\infty} h(x_t)$.

We claim that $\lim_{t\to\infty} \nabla h(x_t) = 0$, hence $M = \min_x h(x)$. We prove the claim by contradiction. Otherwise, there exists $\epsilon_0$ satisfies that for any $T$, $\exists t_0 > T$, $s.t. ||\nabla h(x_{t_0})|| > \epsilon_0$.

Similar to Equation 23,

$$h(x_{t+1}) - h(x_t) \leq \nabla h(x_t)^T (x_{t+1} - x_t) + \frac{L}{2} ||x_{t+1} - x_t||^2$$

$$\leq -\eta \nabla h(x_t)^T \nabla h_t(x_t) + \frac{L\eta^2}{2} ||\nabla h_t(x_t)||^2$$

$$= \eta (\nabla h_t(x_t)^T \nabla h_t(x_t) - \nabla h(x_t)^T \nabla h_t(x_t)) + (\frac{L\eta^2}{2} - \eta) ||\nabla h_t(x_t)||^2$$

$$= \eta (\nabla h_t(x_t) - \nabla h(x_t))^T \nabla h_t(x_t) + -\frac{\eta}{2} ||\nabla h_t(x_t)||^2. \tag{24}$$

In the last equation, we choose $\eta = \frac{1}{L}$. Since $\lim_{t\to\infty} \nabla h_t(x) = \nabla h(x)$, we have $\lim_{t\to\infty} |(\nabla h_t(x_t) - \nabla h(x_t))^T \nabla h_t(x_t)| \leq \lim_{t\to\infty} ||\nabla h_t(x_t) - \nabla h(x_t)|| \cdot ||\nabla h_t(x_t)|| = 0$. Since whatever how large the $T$ is, we can find $t_0 > T$ s.t. $||\nabla h(x_{t_0})|| > \epsilon_0$, so we can find $|h(x_{t_0+1}) - h(x_{t_0})| > \frac{\eta}{4}\epsilon_0$, which is contradict to the convergence of $\{h(x_t)\}_{t=0}^{\infty}$ (Cauchy principle of convergence).

Hence we prove the claim.

Finally, since $h(x)$ is convex, the algorithm will converge to a global minimum of $h(x)$. □

# B  ADDITIONAL EXPERIMENTAL RESULTS

## B.1  ABLATION STUDY

### B.1.1  THE ADAPTIVE DIVERSITY LOSS

We demonstrate the effects of adaptive diversity loss constrained by the population quality described in Section 3.2. The Atari game FishingDerby is used for the ablation study, and three training settings are compared: (1) The baseline setting is the diversity loss function with a fixed coefficient $\lambda_\infty$. (2) In the decay setting, we apply exponential decay of the coefficient $\lambda_\infty$ with decay step $t_0$. (3) In the adaptive setting, we apply an adaptive coefficient $\lambda_j(t, R_{t,j})$ that considers both the relative quality of the agent and the current training steps according to Section 3.2. The hyperparameters for this ablation study are detailed in Table 10.

The experimental results of the ablation study are shown in Figure 4. For the training setting of a fixed coefficient $\lambda^{(0)}$, it is worth noting that the qualities of some agents can get trapped into local optima,

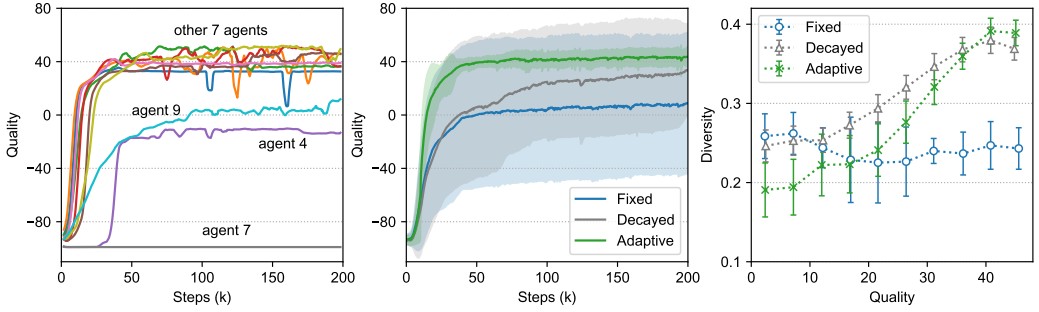

Figure 4: Comparison of three diversity training settings: the fixed diversity loss (fixed), the decayed diversity loss (decayed), and the adaptive diversity loss with quality constraint (adaptive). **Left**: typical training curves of the qualities of a population of 10 agents with the fixed setting. **Middle**: training curves of the qualities of a population of 10 agents for the three settings. **Right**: the QSD performance for the three settings.

as shown in Figure 4 (left), which hinders the optimization of diversity among policies of similar qualities. Moreover, as shown in Figure 4 (middle), the qualities of agents improve slowly for both the fixed and the decay training settings, which may have a negative effect on maximizing diversity among policies of near-optimal qualities, e.g., quality levels above 40 in this game. By introducing the adaptive quality constraint, we observe a much faster improvement of the population quality in Figure 4 (middle). The overall QSD performances of the three settings are shown in Figure 4 (right). We can observe that the quality constraint optimization has slightly better QSD scores at higher quality levels (>60%) although it improves the quality at the cost of diversity degradation at low quality levels. We recommend applying the adaptive diversity loss balanced by the population quality in practice.

We further investigate the effect of the hyperparameter decay rate $t_0$ and initial $\lambda_0$ in the QSD-PBT algorithm. In the Atari game FishingDerby, we first fix the initial $\lambda_0$ to 0.1 and vary the decay rate $t_0$ from 20k to 500k training steps, then fix the decay rate $t_0$ to 500k and vary the initial $\lambda_0$ from 0.05 to 0.4. Ablation results are shown in Figure 5, where computational/time overheads and QSD scores are measured in the same way as in Table 1 and Table 2. From the experimental results, we can conclude that the initial $\lambda_0$ significantly affects the training convergence and demonstrates a trade-off between the final QSD score and the training overhead in a large range (from 0.05 to 0.3). When $\lambda_0$ is too large ($\lambda_0 = 0.4$), QSD-PBT can not reach the optimal quality within an acceptable time overhead (500k steps), then the partial sum of the quality intervals is reported and thereby the QSD score degrades. Compared with $\lambda_0$, QSD-PBT algorithm is less sensitive to the decay rate $t_0$, higher decay rates (20 and 50) lead to similar results as lower initial $\lambda_0$, while a suitable decay rate (200) would both maintain the QSD score and reduce the time overheads.

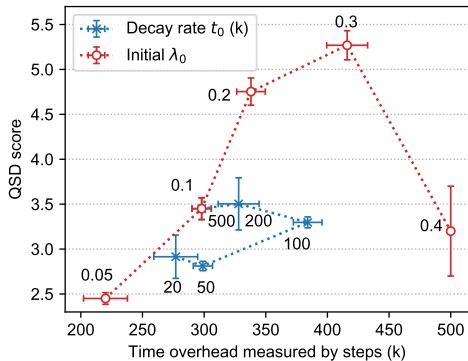

Figure 5: The final QSD scores and the time overheads under different settings of decay rate $t_0$ and initial $\lambda_0$.

### B.1.2 OTHER DIVERSITY MEASURE

In this section, we provide more analysis of the measure function. Note that the measure function is in need in our algorithm to measure the diversity among the policies, so that our algorithm can train to improve the diversity. In the main experiment, we choose the MSE function to be the measure function. We note that our method theoretically can be applied to any measure function as long as the measure function is differentiable. We provide one experiment for example below. However, the choice of the measure function absolutely will affect the performance of the algorithm, and we discuss this later.

We claim that QSD-PBT can be applied to any diversity measure as long as the diversity measure is differentiable with respect to the BDs. We here provide additional experimental results using a different (from the main experiments) diversity measure, i.e., the determinant of DPP, using the MuJoCo task Hopper.

Table 3: The QSD scores on the MuJoCo task Hopper using the determinant of DPP as the diversity measure. #step denotes the number of training steps.

| MuJoCo Task | | EDO-CS | PBT | DvD-TD3 | QSD-PBT (Ours) |
|---|---|---|---|---|---|
| Hopper $R_{max}$: 3k | score > 0% | $0.95 \pm 0.43$ | $0.80 \pm 0.14$ | $2.32 \pm 0.96$ | $\mathbf{9.49 \pm 2.83}$ |
| | score > 60% | $0.10 \pm 0.04$ | $0.09 \pm 0.03$ | $0.31 \pm 0.09$ | $\mathbf{0.45 \pm 0.12}$ |
| | #step (k) | $200 \pm 27$ | $\mathbf{111 \pm 7}$ | $208 \pm 27$ | $170 \pm 10$ |

From the results presented in Table 3, we can conclude that QSD-PBT still outperforms other methods when the diversity measure is the determinant of DPP. The results are consistent with that in Table 1 when the pair-wise distance is employed as the diversity measure. One reason is that QSD-PBT optimizes the user-defined diversity objective directly, while the other methods optimize a task-agnostic diversity measure (PBT, DvD) or quality within different cells of the BD space (EDO-CS). Besides, QSD scores within intervals above 60% of the maximum qualities in Table 3 drop faster compared to the results on the pair-wise distance diversity measure in Table 1. The reason is that the determinant of DPP is much more sensitive to the changes in BDs, in comparison with the mean pair-wise distance measure. Therefore, we recommend the mean pair-wise Euclidean distance as the first choice for users and experiments in this paper.

The choice of measure function will have an impact on the training overheads from two aspects:

(1) The measure function itself may have high computational complexity, e.g., $O(n^3)$ in DPP, where $n$ is the population size. However, $n$ is small in practice (8 in MuJoCo and 10 in Atari) and the complexity is negligible compared to neural network models. For example, when we increase population size n from 10 to 100, we do not observe any decrease of GPU sample speed during training.

(2) Different measure functions will affect the convergence of QSD-PBT and hence affect the computational sources. In Figure 6, we provide training curves of MSE and DPP measure functions in MuJoCo tasks. From the figure, we can see the convergence rate of DPP is slightly slower than MSE. The reason is that MSE calculates the pairwise distance of BDs among all policies in the population and averages them by equal weights, while the DPP is more sensitive to the most similar policies and is unstable. For example, the DPP of a population that contains two same policies in it will be zero, regardless of how diverse the other policies are in the population.

### B.2 RESULTS ON RANDOMLY DESIGNED BDs

Since the trajectory BD can be technically anything, the design principles in this paper are focused on generality and practicality. The BDs in previous experiments include various forms, e.g., action BD (MuJoCo joint torques), state BD (Atari game time), and trajectory BD (Atari left/right preference). However, we find that particular choices of BD still favor certain algorithms. For example, BD defined on actions favors the DvD algorithm since it directly optimizes the KL distance of agents' actions. Hence the improvement of DvD over PBT is more significant in the MuJoCo results (Figure 1) than in the Atari results (Figure 2). Instead of defining meaningful and explicit BDs, we further investigate the performance of QSD-PBT on implicit BDs that would less favor certain algorithms. Since we can

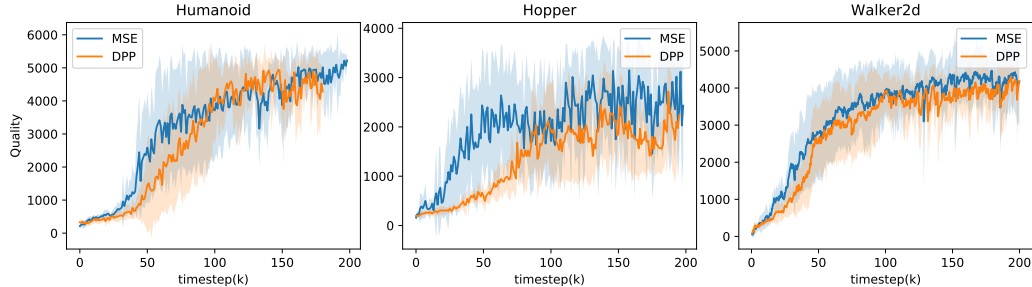

Figure 6: Comparison of the training curve between MSE and DPP measure function. Lines and shadings represent the mean and 1-std range of the qualities among the policies in the population.

only get states as raw pixel images from the Atari environment, we focus on actions and the implicit BD is designed as the randomly weighted action rate:

$$b_i(\tau) = \boldsymbol{w}_i^{\mathrm{T}} \boldsymbol{a}, \ \ \boldsymbol{w}_i \sim \{(w_i^1, ...w_i^K) | \sum_{j=1}^{K} w_i^j = 1; w_i^j > 0, \forall j\} \tag{25}$$

where weight $\boldsymbol{w}_i$ is sampled from the normal distribution and normalized by the *softmax* function, $K = 6$, action rates $\boldsymbol{a}$ is a 6-dimension vector [*fire*, *noop*, *up*, *down*, *left*, *right*], the calculation of action rates is in Table 8. We generate five random BDs, i.e., the weights form a $5 \times 6$ matrix, and provide additional training in the Atari game FishingDerby, the results are shown in Figure 7. Besides, QSD-PBT checkpoints trained on generally designed BDs in Section 4.2 are cross-evaluated by random BDs, marked as "CE". From the experiment, the results are consistent with Figure 1 and Figure 2. QSD-PBT CE is trained with the BD *fire rate* and the diversity of DvD-PPO is defined by action pairs, therefore both of them favor certain random BDs and perform slightly better than PPO and PBT. Comparing QSD-PBT and QSD-PBT CE, we conclude that our algorithm is capable of "what you will get is what you have defined" since it allows for the direct computation of the diversity gradient.

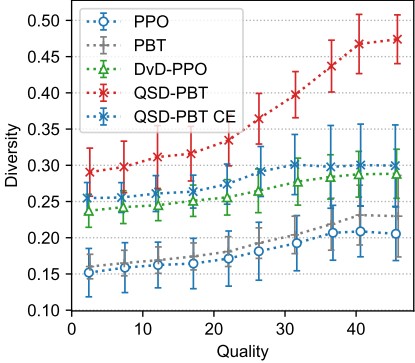

Figure 7: The quality similar diversity across 10 quality intervals on random BDs, "CE" means cross-evaluation on these BDs.

### B.3 DIVERSITY RESULTS ON EACH INDIVIDUAL BD

We claim that QSD-PBT can generate user-intended diversity as long as a diversity measure function and a set of user-defined BDs are provided. We illustrate this ability of QSD-PBT in the main text using orientation BDs in Figure 3. We here provide additional results using the four Atari games for each of the five BDs defined for the Atari experiments.

Results in Table 2 and Figure 2 are further decomposed into each individual BD in Figure 9, where we plot the QSD for a single BD one at a time, i.e., $\mathcal{B}(\pi_\theta) = [B_1(\pi_\theta)]$. After being normalized by

scale factors in Table 8, the five BDs have similar ranges. Overall from Figure 9, we can conclude that QSD-PBT outperforms previous methods in terms of the QSD performance for each of the five BDs defined for the Atari experiments in most cases.

We observe that some user-defined BDs are highly correlated with the quality, e.g., the **game time** in PvE-style games (MsPacMan and RiverRaid), where QSD-PBT achieves only slight improvements compared with other methods. How to adaptively handle the correlation between a BD and the quality remains an interesting question. Besides, for the **left_right** BD in the game RiverRaid, the river course is narrow (shown in supplementary videos), and there is no shortcut (as in the game MsPacMan) that connects the left and right areas. Therefore, left and right actions are restricted to have similar numbers, and the resulting diversity scores are small for all the methods in this game.

### B.4    ADDITIONAL VISUALIZATIONS

We illustrate the diversity at different quality levels in Figure 8 using the Humanoid-v2 task. In this task, an agent can be rewarded by standing or walking, and walking at faster speeds often leads to higher rewards. In Figure 8(a), agents at a lower quality level (4000-5000) learn to stand and balance in various poses, e.g., with open or closed legs and different hand gestures. In contrast, agents at a higher quality level (5000-6000) prefer various walking gaits, including ambling, striding, and mincing, which is demonstrated in Figure 8(b).

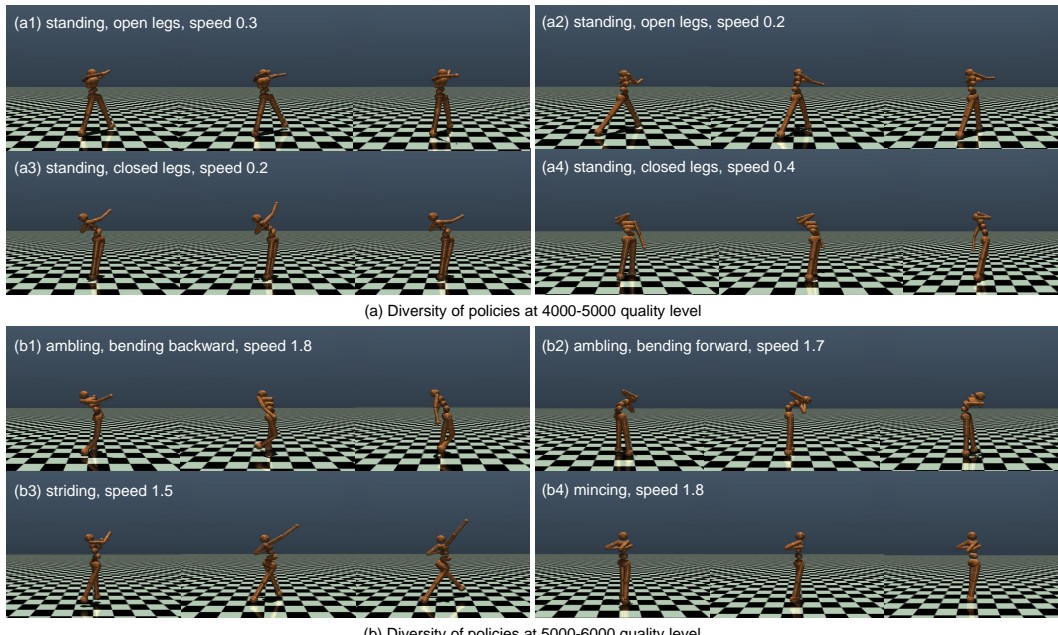

Figure 8: Visualization of diversity at different quality levels on Humanoid-v2.

Additional visualizations of diverse policies generated by QSD-PBT are provided in Atari and MuJoCo videos. The results demonstrate that QSD-PBT is effective in both maximizing diversity across different quality levels and generating user-intended diversity across different user-defined BDs.

For the MuJoCo videos, we focus on demonstrating the diversity at different quality levels. For the Humanoid task, agents at a high quality level (5000-6000) prefer various walking gaits, e.g., ambling, striding, and mincing. Agents at a low quality level (4000-5000) only learn to stand still and balance in various poses. For the Walker2d task, agents at a high quality level (5000-6000) learn to run or stride, while agents at a low quality level (4000-5000) can only walk with small steps or just hop. For the Hopper task, agents at a high quality level (3000-4000) prefer fast hopping with different poses, while agents at a low quality level (2000-3000) only learn to stand or jump with small steps.

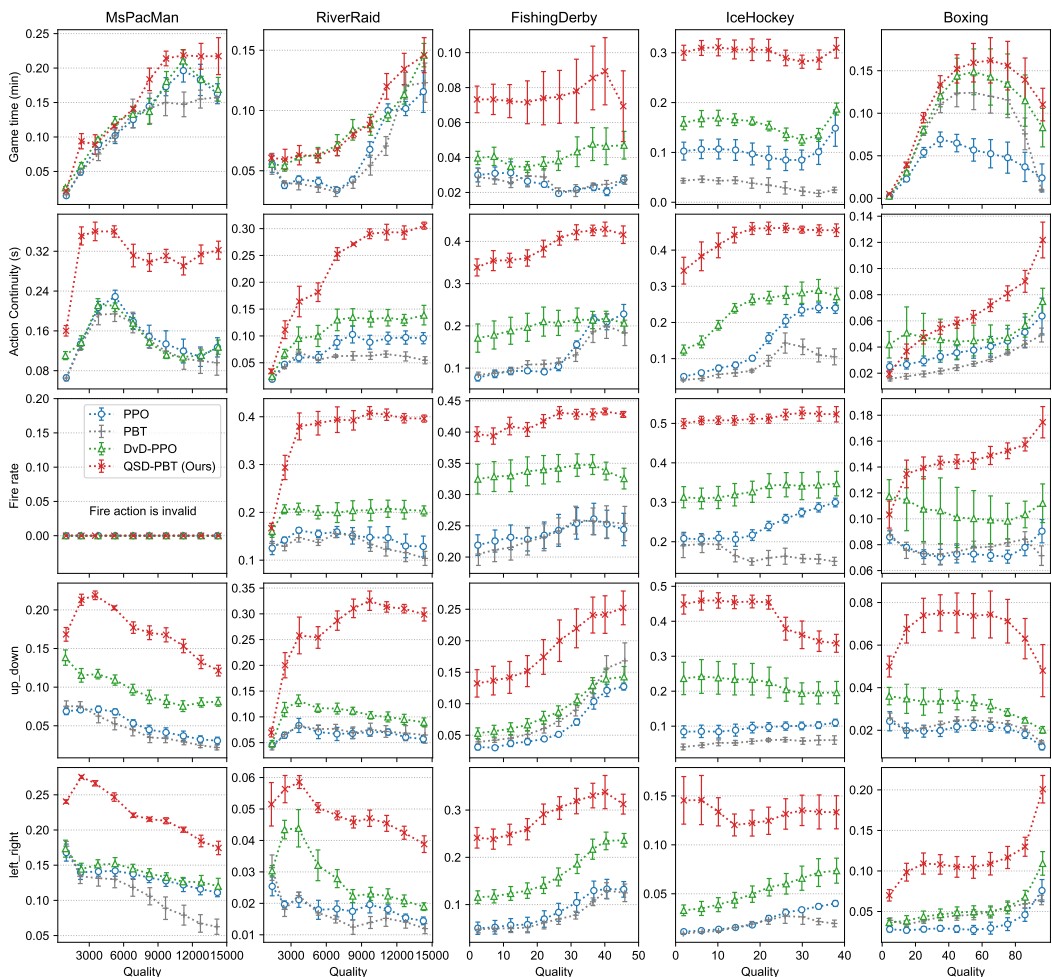

Figure 9: Detailed diversity results for individual BDs in Atari games

For the Atari videos, we focus on demonstrating the diversity of the five user-defined BDs. Aside from a video version of Figure 3 in MsPacMan, the orientation BDs (**left_right** and **up_down**) are further visualized in FishingDerby, where PPO agents always fish in the up and left area since it is an optimal and easy-to-explore solution. The diversity on the other three BDs (**game time**, **fire rate**, and **action continuity**) are demonstrated in IceHockey and RiverRaid. In the game IceHockey, agents get scores by various strategies, and three of them are displayed: (1) Dribble, run, and shoot (fire) in the normal style. (2) Stand still, defend, and shoot far away. (3) Swing left and right without shooting. In the game RiverRaid, agents can speed up or slow down by up or down actions, and therefore the total game time differs. Our agents can generate diverse behaviors by considering multiple aspects during driving, e.g., the speed, obstacles, and the remaining fuel.

## B.5 Trade-off between quality and diversity

Generally speaking, the quality and the diversity are a trade-off, higher quality tends to result in lower diversity. But as shown in Figure 1, Figure 2 and Figure 5, since environments and BDs are so varied, it is hard to clarify the accurate correlation at each quality level and we need to analyze it case-by-case. Usually it depends on (1) the correlation of the BDs and the quality, and (2) the exploration space and complexity of the environment. Here we observe the difference between Figure 1 and Figure 2. In MuJoCo tasks, the QSD curve shows the quality and diversity trade-off, while in Atari games, the trade-off disappears.

We conclude with two major reasons: (1) The definition of BDs. For MuJoCo tasks, we define most of the BDs on joint torques, which are highly correlated to the pose control of the robots and the resulting score. While for Atari games, we define relatively coarse-grained BDs that have lower correlations with the quality, e.g., the left or right preference in Figure 3 does not influence the score since most games are mirror symmetric. (2) The exploration space and complexity of the environment. Compared with MuJoCo tasks, Atari games are more complex and have more exploration space. Take the game RiverRaid for example, at a lower quality level (<3000), the initial river course is narrow and the agent can only go straight. When it reaches certain scores, the agent will enter into new scenes that have a much broader course. The scene switching provides more exploration and diversity space, resulting in the improvement of diversity along with quality.

## C    IMPLEMENTATION DETAILS

### C.1    DESCRIPTIONS OF THE BDs

For MuJoCo tasks, the BDs are defined on the scoring speed and the built-in joint torques respectively. The joint torques are the actions in MuJoCo tasks. The action spaces for each MuJoCo task are shown in Table 4-6, and more details can be found here. For a trajectory, we divide its total score by the number of frames used to obtain the BD of **Scoring speed**. For each joint in a trajectory, we sum the actions, i.e., the torques, applied to it and divide the sum by the number of frames to obtain a joint BD. As a result, the number of BDs for each MuJoCo task is the number of joints in that task plus one, i.e., the **Scoring speed**. We normalize each BD to the range of $[0, 1.0]$. An overview of all the BDs for each task in the MuJoCo experiment is presented in Table 7.

Table 4:  Action space for Hopper.

| Num | Description (Torque applied to different joints) | Range |
|-----|--------------------------------------------------|-------|
| 1   | the thigh rotor                                  |       |
| 2   | the leg rotor                                    | $[-1, 1]$ |
| 3   | the foot rotor                                   |       |

Table 5:  Action space for Humanoid.

| Num | Description (Torque applied to different joints) | Range |
|-----|--------------------------------------------------|-------|
| 1   | the hinge in the y-coordinate of the abdomen     |       |
| 2   | the hinge in the z-coordinate of the abdomen     | $[-0.4, 0.4]$ |
| 3   | the hinge in the x-coordinate of the abdomen     |       |
| 4   | the rotor between torso/abdomen and the right hip (x-coordinate) |       |
| 5   | the rotor between torso/abdomen and the right hip (z-coordinate) |       |
| 6   | the rotor between torso/abdomen and the right hip (y-coordinate) |       |
| 7   | the rotor between the right hip/thigh and the right shin |       |
| 8   | the rotor between torso/abdomen and the left hip (x-coordinate) |       |
| 9   | the rotor between torso/abdomen and the left hip (z-coordinate) |       |
| 10  | the rotor between torso/abdomen and the left hip (y-coordinate) |       |
| 11  | the rotor between the left hip/thigh and the left shin | $[-0.4, 0.4]$ |
| 12  | the rotor between the torso and right upper arm (coordinate -1) |       |
| 13  | the rotor between the torso and right upper arm (coordinate -2) |       |
| 14  | the rotor between the right upper arm and right lower arm |       |
| 15  | the rotor between the torso and left upper arm (coordinate -1) |       |
| 16  | the rotor between the torso and left upper arm (coordinate -2) |       |
| 17  | the rotor between the left upper arm and left lower arm |       |

For Atari games, we define a common set of BDs for each game. There are 5 different BDs in total. The design of these BDs are motivated by covering different ways in which diverse and meaningful

Table 6: Action space for Walker2d.

| Num | Description (Torque applied to different joints) | Range |
|-----|--------------------------------------------------|-------|
| 1 | the right thigh rotor | |
| 2 | the right leg rotor | |
| 3 | the right foot rotor | |
| 4 | the left thigh rotor | $[-1, 1]$ |
| 5 | the left leg rotor | |
| 6 | the left foot rotor | |

Table 7: An overview of all BDs for each MuJoCo task. #frame is the total number of frames in a trajectory.

| BD | Definition | Normalization |
|----|-----------|---------------|
| Scoring speed | $\frac{\text{total\_score}}{\#\text{frame}}$ | $\frac{1}{5}x$ |
| Joint torques in Hopper ($\dim = 3$) | $\frac{\sum actions}{\#\text{frame}}$ | $\frac{x+1}{2}$ |
| Joint torques in Humanoid ($\dim = 17$) | $\frac{\sum actions}{\#\text{frame}}$ | $\frac{x+0.4}{0.8}$ |
| Joint torques in Walker2d ($\dim = 6$) | $\frac{\sum actions}{\#\text{frame}}$ | $\frac{x+1}{2}$ |

policies could possibly differ. We normalize each BD to a similar range. An overview of all the BDs in the Atari experiment is presented Table 8.

Table 8: An overview of all BDs for Atari games. # indicates the total number of frames, action changes, or specific actions in a trajectory.

| BD | Definition | Scale factor |
|----|-----------|--------------|
| Game time (minute) | $\frac{\#\text{frame}}{900}$ | $\frac{1}{3}$ |
| Fire rate (per second) | $\frac{15 \cdot \#\text{fire}}{\#\text{frame}}$ | $\frac{1}{3}$ |
| Action continuity | $\log\left[\frac{15 \cdot \#\text{action\_change}}{\#\text{frame}}\right]$ | $\frac{1}{15}$ |
| Left_right | $\log\left[\frac{\#\text{left}}{\max(\#\text{right},1)}\right]$ | $\frac{1}{15}$ |
| Up_down | $\log\left[\frac{\#\text{up}}{\max(\#\text{down},1)}\right]$ | $\frac{1}{15}$ |

## C.2 THE DIVERSITY MEASURES

We use the mean pair-wise distance as the diversity measure for the main experimental results. For a population of policies $\Pi$ with size N, $\Pi = \{\pi_{\theta_j} | 1 \leq j \leq N\}$, the mean pair-wise distance is defined as:

$$\frac{2}{N(N-1)} \sum_{i=1}^{N-1} \sum_{j=i+1}^{N} ||\mathcal{B}(\pi_{\theta_i}) - \mathcal{B}(\pi_{\theta_j})||_2, \tag{26}$$

where $\mathcal{B}(\pi_\theta)$ is the vector of all the BD values of a policy defined in Section 2.

As we stated in the main text, our method QSD-PBT can be applied to any explicit diversity measure as long as the measure is differentiable with respect to $\mathcal{B}(\pi_\theta)$. Later in Appendix B.1.2, we present results using another diversity measure, i.e., the determinant of a DPP. The DPP determinant of $\Pi = \{\pi_{\theta_j} | 1 \leq j \leq N\}$ is defined as:

$$\det[\mathcal{K}(\mathcal{B}(\pi_{\theta_i}), \mathcal{B}(\pi_{\theta_j}))_{i,j=1}^{N}], \tag{27}$$

where $\mathcal{K}$ is a given kernel function. We set $\mathcal{K} = \exp[-\frac{||\mathcal{B}(\pi_{\theta_i}) - \mathcal{B}(\pi_{\theta_j})||_1}{2}]$ in our case.

### C.3 The Calculation of the QSD Score

In practice, the number of policies obtained throughout training with qualities lying in the same interval can be much larger than $N$, which is the population size of $\Pi$ used for evaluating the diversity measure $\mathrm{Div}(\Pi)$. To make an efficient and fair comparison across different methods in terms of the QSD score, for each quality interval we sample a population $\Pi$ of size $N$ for 100 times. The way we sample $\Pi$ is by alternating the training index of the policy till $N$ policies are obtained, because policies with the same training index tend to have similar BD values. The training index of a policy denotes which agent in the population the policy comes from during population-based training. We calculate the diversity measure for each sampled $\Pi$ and use the sample mean of the 100 evaluations as the diversity measure for the corresponding quality interval, which is then aggregated according to Equation 1 to produce the QSD score.

### C.4 Hyperparameters and Details of the Compared Baseline Methods

For MuJoCo tasks, we compare QSD-PBT to three population-based training algorithms: EDO-CS, PBT, and DvD. EDO-CS uses the ES for optimizing the quality and a selection mechanism to induce diversity among the defined BDs. PBT uses TD3 as the backbone and mostly focuses on the quality of each policy in the population. The *Perturb* exploration and *Truncation selection* exploitation strategies are adopted every 10k training steps for our setting of PBT. The DvD-TD3 in the original paper is implemented. DvD-TD3 optimizes a combined loss of quality and a task-agnostic diversity measure defined on state action probabilities. All the hyperparameters for each method are listed in Table 9 except for EDO-CS and QD-PG, for which we implement with the architecture and hyperparameters suggested in their papers. For QSD-PBT, we use the LSTM as a feature exactor in BD estimators. We pass the trajectory state and trajectory action to an LSTM, respectively, then concatenate the feature and pass it to the MLP. The 'state' BD we used for reproducing QD-PG is the action (a real-valued vector, which has a dimension of 17, 3, and 6 respectively in the three MuJoCo tasks) in a state. We believe this is a fair setting for QD-PG, because the trajectory BD we defined here is the average torques (i.e., actions) applied to the hinge joints over a trajectory.

For Atari games, we replace the baseline method EDO-CS with PPO for training efficiency consideration, where a population of $N$ independent PPO agents are trained. As Atari games have discrete actions and high-dimensional image inputs, all the methods employ the DQN (Mnih et al., 2015) architecture as the backbone model. We find it effective to provide statistics from the beginning of a game to the current state (the number of frames, the number of fire, left, right, up, down actions and the number of action changes) as a sufficient encoding of trajectory $\tau_{0:j}$. Instead of the LSTM applied in MuJoCo experiments, these additional features are applied in the DQN model to better estimate BD defined in Table 8. Accordingly, we use PPO in the implementation of both DvD and PBT. All the hyperparameters for each method are listed in Table 10.

### C.5 Pseudocode of QSD-PBT

The developed QSD-PBT is in general a population-based RL algorithm. In QSD-PBT, we employ parallel actors and learners to simultaneously train a population of $N$ agents. The policy of each agent is trained using the gradient of the loss function defined in Equation 9. Meanwhile, the policy value functions or Q functions are trained using the mean squared error, and so are the state BD estimators or the state-action BD estimators of each agent. Moreover, QSD-PBT maintains a running average estimation of each agent's quality, each agent's BDs, and the population's mean quality. The pseudocode of QSD-PBT is given in Algorithm 1. Note that, without loss of clarity, we present QSD-PBT with TD3 and QSD-PBT with PPO in one algorithm.

Table 9: Hyperparameters for MuJoCo tasks.

| Hyperparameters | Value |
|---|---|
| *Shared* | |
| Population size ($N$) | 8 |
| Replay buffer size | $10^6 \times N$ |
| Mini-batch size | $1024 \times N$ |
| Optimizer | Adam |
| Learning rate | 3e-4 |
| Discount factor ($\gamma$) | 0.99 |
| Target network update rate ($\tau$) | 0.005 |
| Std of gaussian exploration noise | 0.1 |
| Delayed policy update | 2 |
| Policy noise | 0.1 |
| Noise clip | 0.5 |
| Policy network | MLP (state_dim-256-256-action_dim) |
| Critic network | MLP (state_dim+action_dim-256-256-1) |
| BD Estimator | LSTM(128) + MLP (256-256-BD_dim) |
| *PBT* | |
| Exploitation interval | every 10k steps |
| Truncation selection ratio | 0.25 |
| *DvD-TD3* | |
| DvD loss factor | 0.5 |
| DvD embedding mini-batch size | 256 |
| *QSD-PBT* | |
| Adaptive diversity loss $\lambda^{(0)}$ | 0.1 (Hopper, Humanoid), 0.5 (Walker2d) |
| Adaptive diversity loss $t_0$ | $2 \times 10^4$ |

---

**Algorithm 1:** QSD-PBT with TD3 or PPO baseline

---

Initialize network parameters for a population of $N$ agents: $\{\theta_1, \theta_2, ..., \theta_N\}$.
Start multiple actors and learners in parallel.
**Actors**: **while** *true* **do**
    Fetch the latest model from the learners, and run multiple game episodes.
    Generate RL training samples in the form: $x = [a_t, s_t, s_{t+1}, r_t]$, and the encoding result of
      $\tau_{0:t-1}$, and send them to the replay buffer.
    For each agent, calculate the average BD results $\hat{\mathcal{B}}(\pi_\theta)$ and the quality $R$ using results in
      multiple games, and send $R, \mathcal{B}(\pi_\theta), \hat{\mathcal{B}}(\pi_\theta)$ to a monitor.
**Learners**:
**for** $t \in 1, 2, 3, ...$ **do**
    Fetch current BD results $\mathcal{B}(\pi_\theta), \hat{\mathcal{B}}(\pi_\theta)$, and qualities $R$ of all agents from the monitor.
    Calculate the diversity measure function $\mathrm{Div}(\Pi)$.
    **for** $n \in 1, 2, ..., N$ **do**
        Calculate the diversity gradient of $\mathrm{Div}(\Pi)$ at $\hat{\mathcal{B}}$.
        Fetch a batch of samples for agent $n$ from the replay buffer.
        Calculate value loss and policy loss according to TD3 or PPO algorithm.
        Calculate mean-square loss for BD estimators $V_B^{\pi_{\theta_n}}$ by $\mathcal{B}(\pi_{\theta_n})$.
        Calculate diversity loss according to Equations 6, 8.
        Update critic parameters $\theta_n$ using gradients on value loss and BD estimator loss.
        Update actor parameters $\theta_n$ using gradients on $L_{total}(\pi_{\theta_n})$ as in Equation 9.

---

Table 10: Hyperparameters for Atari games.

| Hyperparameters | Value |
| --- | --- |
| *Shared (PPO)* | |
|     Population size ($N$) | 10 |
|     Number of actors | 700 (CPUs) |
|     Number of learners | 8 (GPUs) |
|     Frame skip | 4 |
|     Reward shaping | $\log(\mathrm{abs}(r)+1) \cdot (2 \cdot \mathbf{1}_{\{r \geq 0\}} - \mathbf{1}_{\{r<0\}})(Fanet\ al.,\ 2021)$ |
|     Replay buffer size | $50000 \times N$ |
|     Mini-batch size | $256 \times N$ |
|     Optimizer | Adam |
|     Learning rate | 1e-4 |
|     Discount factor ($\gamma$) | 0.99 |
|     GAE parameter ($\lambda$) | 0.95 |
|     PPO clipping ratio | 0.2 |
|     Value function coefficient $c_1$ | 0.5 |
| *PBT* | |
|     Exploitation interval | every 20k steps |
|     Truncation selection ratio | 0.2 |
| *DvD-PPO* | |
|     DvD loss factor | 0.1 |
|     DvD embedding mini-batch size | 512 |
| *QSD-PBT* | |
|     Adaptive diversity loss $\lambda^{(0)}$ | 0.1 |
|     Adaptive diversity loss $t_0$ | $5 \times 10^5$ |

