# OpenReview forum: "Quality-Similar Diversity via Population Based Reinforcement Learning"
_ICLR.cc/2023/Conference — ICLR 2023 poster_

### Official Review · Reviewer_bTHx · 2022-10-16

**Confidence:** 3
**Correctness:** 3
**Technical Novelty And Significance:** 3
**Empirical Novelty And Significance:** 2
**Recommendation:** 6

**Clarity, Quality, Novelty And Reproducibility:**

Clarity: The paper is *relatively* clear and easy to follow. However, there are points, which pauses/baffles the reader (see the comments below).

Quality: The submission provides both theoretical exposition of the problem and experimental evaluation of the proposed method. Broadly speaking, it meets the level of an accepted ICLR paper.

Reproducibility: The authors provide several hyper parameters and detailed general implementations. However, I do not see a reproducibility statement in the paper about code release.

**Strength And Weaknesses:**

Strengths:

1. The main objective (to produce a set of diverse policies at multiple quality levels) seems useful and interesting to investigate.
2. Formulation of the Quality-Similar Diversity (QSD) is novel and seems to achieve superior results.

Weaknesses:

1. The paper lacks some analysis of sensitivity as well as computational overhead. In particular, it is not clear how the rate with which $\lambda_i$'s are changed will impact convergence and stability of the algorithm as a whole. Additionally, the choice of $f$ can have significant impact on wall-clock time and computational requirements.
2. The exposition of concepts can be improved as it lacks accuracy at points, which causes confusion (see the comments below).

**Summary Of The Paper:**

This paper addresses task-specific diversity problem, which exhibits in policies with similar expected return (what the authors called quality). The task specificity is framed by a set of scalar functions of a trajectory defined by the user, which are called Behavior Descriptors (BDs). A BD captures some property of a given trajectory, which is preferred by the user, e.g., number of times that a self-driving agent shifts gear (say the less the better). The authors propose an estimator for each BD, based on which they compute the gradient of BD w.r.t the policy's parameters. This gradient is then used to augment standard policy grad in a tunable manner, the result of which is a class of algorithms, called population-based RL algs.

**Summary Of The Review:**

- Definition of BD needs more explanation. Do you need some restrictions such as being bounded (or finite)?

- First paragraph of page 3: what do you mean by a measure function? Do you mean a function satisfying to be non-negative, zero for the empty set, and countable additive? If yes, I would probably make it explicit and simply discuss what these properties mean in the case of $f$. If no, then the definition of $f$ does not require it to be non-negative. I am wondering if $f$ can also be negative, then the QSD score can be zero for some given M-partition even if all $\mathcal{B}_i$ are strictly non-zero. This may make the definition counterintuitive.

- Page 3, first paragraph: depending on the definition of $f$, its evaluation might become a computational bottleneck. Some explanation on computational complexity would be apropos.

- Section 3, Lines 5-6: “Note that both the user-specified BD and the quality of a policy are defined on a trajectory” -> not correct according to your definition of quality. The return (not the quality) is defined for a trajectory.

- Page 4, first paragraph: suggestion: for the BD advantage function, you can observe that its expectation over $\pi_{\theta}$ is zero, which simply follows from your definition of value functions. Hence, in practice, it might be quite beneficial to subtract the mean (taken over mini-batch) from BD advantage to make the implementation more robust.

- Moreover, for computation of advantage functions, I strongly recommend the following rather than GAE:
“Direct Advantage Estimation”, https://arxiv.org/abs/2109.06093

- Section 3.2: For maximizing diversity, section 4 of the following reference also seems relevant:
https://arxiv.org/abs/2110.01548
(They have an ensemble of Q networks, while you have a set of policy networks.)


Minors:

- Last paragraph of Introduction: the acronym PBT used before it is defined.

- First paragraph of section 2: your reward function is indeed an expected reward as it only defines over state-action and not the full transition ($r(s,a) = \mathbb{E}_{s’} ~r(s,a,s’)$).

- First paragraph of section 2: $T$ is not defined. More importantly, it looks like that you are considering episodic MDPs (also evident from allowing $\gamma=1$). If so, make it explicit. Moreover, it is not clear whether you have termination at $t=T$ or not. If yes, you need to also define terminal states (whose value is zero by definition).

- Section 3, 3rd line -> I would say “conventional” rater than “traditional”, since the reference is too recent to be a tradition.

---

> ### Author Response · Authors · 2022-11-17
> **Response to Reviewer bTHx (Part 2)**
>
> ***
>
> > **[Q4]** First paragraph of page 3: what do you mean by a measure function? Do you mean a function satisfying to be non-negative, zero for the empty set, and countable additive? If yes, I would probably make it explicit and simply discuss what these properties mean in the case of $f$...
>
> **[A4]** Good question. Since BD values of a policy form a multidimensional vector, we need a function to measure the overall diversity of the population, which is called the diversity measure function in this paper.
>
> $f$ should follow these properties:
>
> 1. $f$ should be bounded and non-negative for easy implementation;
>
> 2. Since we do not define the order of an agent in the population, $f$ should be invariant of any permutation of the policies, e.g., $f(b_1, b_2, b_3) = f(b_2, b_1, b_3)$;
>
> 2. $f$ should be differentiable so that we can derive its gradient.
>
> Besides, there are not many diversity measure functions (the two most common ones: MSE and DPP) studied in previous related works. Although the QSD-PBT algorithm allows the user to specify a new measure function, the evaluation in this paper focuses only on the MSE and DPP measure functions.
>
> Thanks for your suggestion. We have made it explicit in the revision, Section 2.
>
> ***
>
> > **[Q5]** Section 3, Lines 5-6: “Note that both the user-specified BD and the quality of a policy are defined on a trajectory” -> not correct according to your definition of quality. The return (not the quality) is defined for a trajectory.
>
> **[A5]** Thanks for the suggestion. The return is defined on the trajectory, and the quality is defined on the policy. We have made it clearer in the revision, Section 2 and Section 3.
>
> ***
>
> > **[Q6]** Page 4, first paragraph: suggestion: for the BD advantage function, you can observe that its expectation over is zero, which simply follows from your definition of value functions. Hence, in practice, it might be quite beneficial to subtract the mean (taken over mini-batch) from BD advantage to make the implementation more robust.
>
> **[A6]** Thanks for the suggestion. Actually, in the implementation of PPO backbone in Atari experiments, we used the advantage norm. Please refer to https://github.com/openai/baselines/blob/master/baselines/ppo2/model.py#L139.
>
> We have made it clear in the revision Section 4.
>
> ***
>
> > **[Q7]** Moreover, for computation of advantage functions, I strongly recommend the following rather than GAE: “Direct Advantage Estimation”, https://arxiv.org/abs/2109.06093
>
> **[A7]** Thanks for the suggestion. It looks interesting to adapt the DAE into BD advantage estimations, and we believe our algorithm can benefit from it. Currently, since we focus on the QSD problem and the diversity gradient computing method, we simply applied the conventional GAE method.
> We have added the discussion in the revision, Section 3.1.
>
> ***
>
> > **[Q8]** Section 3.2: For maximizing diversity, section 4 of the following reference also seems relevant: https://arxiv.org/abs/2110.01548 (They have an ensemble of Q networks, while you have a set of policy networks.)
>
> **[A8]** Thanks for your suggestion, we have added the citation and discussion in the revision, Section 6.
>
> Generally, the population-based model can be implemented either by building an independent model for each population agent, or sharing most of the feature extraction part, e.g., the 3-layer convolution model in DQN and building an ensemble of policy and value heads for each population agent. The latter implementation is similar to the reference.
>
> ***
>
> > **[Q9]** Last paragraph of Introduction: the acronym PBT used before it is defined.
>
> **[A9]** We have corrected it in the revision, Section 1.
>
> ***
>
> > **[Q10]** First paragraph of section 2: your reward function is indeed an expected reward as it only defines over state-action and not the full transition.
>
> **[A10]** Thanks for your suggestion. The reward function in this paper follows the conventional definition in RL, and our reward is indeed the expected reward function.
>
> We have made it clear in the revision, Section 2.
>
> ***
>
> > **[Q11]** First paragraph of section 2:  T is not defined. More importantly, it looks like that you are considering episodic MDPs (also evident from allowing ). If so, make it explicit. Moreover, it is not clear whether you have termination at t=T or not. If yes, you need to also define terminal states (whose value is zero by definition).
>
> **[A11]** Thanks for your suggestion. We consider the episodic MDPs. $T$ is the terminal step, and the state value of the terminal state equals to $0$.
>
> We have made it explicit and clear in the revision, Section 2.
>
> ***
>
> > **[Q12]** Section 3, 3rd line -> I would say “conventional” rater than “traditional”, since the reference is too recent to be a tradition.
>
> **[A12]** Thanks for your suggestion, we have changed the word in the revision.
>
> ***

---

> > ### Comment · Reviewer_bTHx · 2022-12-06
> > **Discussion**
> >
> > Thanks the authors for the detailed replies and additions. I am positive about this paper and would support acceptance.

---

> ### Author Response · Authors · 2022-11-17
> **Response to Reviewer bTHx (Part 1)**
>
>
> ***
>
> > **[Q1]** The paper lacks some analysis of sensitivity as well as computational overhead. In particular, it is not clear how the rate with which $\lambda$'s are changed will impact convergence and stability of the algorithm as a whole.
>
> **[A1]** Thanks for your suggestion, for the computational overhead, we have evaluated the time steps needed for QSD-PBT to reach the optimal quality in Table 1 and Table 2, please refer to the "#step(k)" rows.
>
> We are aware that the overheads are evaluated with only one set of hyperparameters.
> Therefore, we provide additional experiments by varying the decay rate $t_0$ and the initial value of lambda $\lambda_0$ to show how the hyperparameter will impact the convergence and stability of the algorithm. The results are shown in the revision Appendix B.2.1, also shown in https://ibb.co/MnJK3pg.
>
> From the experimental results, we can conclude that:
>
> 1. The initial $\lambda_0$ significantly affects the training convergence and clearly demonstrates a trade-off between the final QSD-score and the training overhead in a large range (from $0.05$ to $0.3$).
> When $\lambda_0$ is too large ($0.4$), QSD-PBT can not reach the optimal quality within an acceptable time overhead (500k steps). The partial sum of the quality intervals is reported, and thereby the QSD-score degrades.
>
> 2. Compared with $\lambda_0$, QSD-PBT algorithm is less sensitive to the decay rate $t_0$. Higher decay rates ($20$ and $50$) lead to similar results as lower initial $\lambda_0$, while a suitable decay rate ($200$) would both maintain the QSD-score and reduce the time overheads.
>
> As for the overhead of population-based training, it can be alleviated by sharing parameters across multiple agents.  We have added the discussion in the revision, Section 6.
>
> ***
>
> > **[Q2]** Additionally, the choice of $f$ can have significant impact on wall-clock time and computational requirements ... Page 3, first paragraph: depending on the definition of $f$, its evaluation might become a computational bottleneck. Some explanation on computational complexity would be apropos.
>
> **[A2]**
> We provide additional training curves of MSE and DPP measure functions in MuJoCo benchmark (https://ibb.co/9t2ZQ8f), and add more discussion in the revision, Appendix B.2.
>
> The choice of measure function $f$ may affect the training overheads from two aspects:
>
> 1. The measure function itself may have high computational complexity, e.g.,  $O(n^3)$ in DPP, where n is the population size. However, $n$ is small in practice ($8$ in MuJoCo and $10$ in Atari) and the computational time of $f$ is negligible compared to neural network models. For example, when we increase population size $n$ from $10$ to $100$, we do not notice any decrease of GPU sample speed during training.
>
> 2. Different measure functions will affect the convergence of QSD-PBT and hence affect the overall training overhead. The convergence rate of DPP is slightly slower than MSE. The reason is that MSE calculates the averaged pairwise distance of BDs among all policies in the population, while the DPP somewhat focuses more on the most similar policies and is more unstable. For example, the DPP of a population that contains two same policies in it will be zero, regardless of how diverse the other policies are in the population.
>
> ***
>
> > **[Q3]** Definition of BD needs more explanation. Do you need some restrictions such as being bounded (or finite)?
>
> **[A3]** Thanks for your suggestion. We have added explanations and examples of BDs in Section 2.
>
> Our definition of BD is a general and flexible form, which only requires that BD is an evaluation function of a trajectory that is finite and easy to be implemented.
>
> For instance, a BD could be the ratio between left and right movements in the trajectory of Atari MsPacMan, or the farming speed (average gold per minute) of a player in MOBA games.
>
> The trajectory BD is a general form and can be simplified to state or action BD when the user is only interested in certain states or actions in a trajectory. For instance, the terminal position in MsPacMan, or the final KDA (kill, death, assistant) statistics in MOBA games.
>
> ***

---

### Official Review · Reviewer_7pVo · 2022-10-24

**Confidence:** 3
**Correctness:** 3
**Technical Novelty And Significance:** 3
**Empirical Novelty And Significance:** 3
**Recommendation:** 8

**Clarity, Quality, Novelty And Reproducibility:**

The paper is written clearly and the quality and novelty of technical contributions are good. Code is not included in the supplementary material.

**Strength And Weaknesses:**

**Strengths**
- The problem statement is interesting and distinct from diversity problems studied in prior work. In particular, this paper aims at finding a set of diverse policies in a hierarchy of quality levels. This is useful in certain AI applications such as games and appears related to curriculum learning tasks. The problem formulation in this paper can be of independent interest to other topics in RL as well.
- Besides the quality-similar diversity problem statement, another technical contribution of this paper is noting that an exact gradient of the user-specified BD based on trajectories can be computed directly using the policy gradient theorem. Thus, one can obtain an unbiased sample-based approximation to the gradient and utilize techniques developed for policy gradient methods.
- Guidelines for adaptive diversity loss and handling multiple quality levels are useful.
- The paper presents a thorough review and categorization of prior diversity methods.
- The paper is well-written and well-organized. The authors provide a solid motivation behind their work and discuss applications and intuition behind their design choices throughout the paper.
- The empirical evaluations show strong performance compared to prior methods in both continuous and discrete tasks.

**Weaknesses**

- The setting considered in the paper requires multiple design choices, which brings the practicality of the method into question. Apart from designing rewards, one needs to design a set of diversity design functions that depend on trajectories (which include a large number of states and actions) and may be difficult to elicit desired behavior. Another design element is the diversity measure function (an example is given which is the determinant of a kernel matrix.)
- In Section 3.2, the authors mention a relationship between large $\lambda$, which determines the strength of diversity, and optimistic initial values used for exploration. This connection is unclear and the two seem rather distinct from my perspective.
- Despite the authors discussing this in Section 4 and Appendix, It is still unclear to me whether particular choices of BD would favor certain algorithms. Since BDs can be technically anything, it would be good to also empirically compare the performance of algorithms given random BDs.


**Summary Of The Paper:**

The paper studies the topic of diversity in reinforcement learning. The authors propose the quality similar diversity problem, whose goal is finding diverse policies within multiple quality levels. In particular, the paper considers task-specific diversity where the user defines certain behavior descriptors (BDs) that are a function of trajectories and capture the type of diversity the user prefers. Given this problem statement, the authors present methods for estimating BDs via the policy gradient theorem. The gradient is then combined with a population-based training via an adaptive diversity loss. This leads to an RL algorithm that optimizes the quality-similar population diversity of a set of policies. Empirical evaluations show improved performance over prior methods w.r.t. diversity of policies across multiple quality levels.

**Summary Of The Review:**

The presented method appears to be useful, practical, simple, and well-motivated. The authors do a good job explaining their method, justifying their design choices, and providing intuition. There are no major weaknesses that would justify rejection.

---

> ### Author Response · Authors · 2022-11-17
> **Response to Reviewer 7pVo (Part 2)**
>
> ***
>
> > **[Q4]** Despite the authors discussing this in Section 4 and Appendix, It is still unclear to me whether particular choices of BD would favor certain algorithms. Since BDs can be technically anything, it would be good to also empirically compare the performance of algorithms given random BDs.
>
> **[A4]** We agree with the reviewer that particular choices of BD would favor certain algorithms. For example, BD defined on action would favor the DvD algorithm since it cares only the KL distance of agents' actions. As a result, the improvement of DvD over PBT is more significant in MuJoCo results (Figure 1) than that in Atari results (Figure 4).
>
> Since BDs can be technically anything, the design principles of BDs in this paper are focused on generality and practicality, i.e., we prefer meaningful yet necessarily biased (would favor certain algorithms) BDs rather than implicit/abstract/unbiased BDs.
>
> Besides, the BDs in our experiments include various forms, e.g., action BDs (MuJoCo joint torques), state BDs (Atari game time), and trajectory BDs (Atari
> left/right preference). QSD-PBT constantly gets the best performance in most of the experiments, since it allows for user-intended BDs and the direct computation of the diversity gradient.
>
> We have made it more clear in Section 4.
>
> Designing some form of random BDs to exclude the favor of certain algorithms is indeed worth investigating. We provide additional experiments on Atari game FishingDerby, and the random BD is designed as the randomly weighted action rate:
>
> $b_i(\tau) = {w}_i^{\rm T} {a}$,
>
> and
>
> $w_i \sim  $  \{ $ \{ (w_i^1,...w_i^K)|\sum_{j=1}^K w_i^j=1; w_i^j>0, \forall j \} $  \}
>
> where weight ${w}_i$ is sampled from the normal distribution and normalized by the softmax function, $K=6$, action rates ${a}$ is a 6-dimension vector [fire, noop, up, down, left, right].
>
> We generate five random BDs, i.e., the weights form a $5 \times 6 $ matrix, and provide additional training in the Atari game FishingDerby. The results are shown in the revision Figure 9 and in https://ibb.co/D1BKN73.
> Besides, QSD-PBT checkpoints trained on generally designed BDs in Section 4.2 are cross-evaluated by random BDs, marked as "CE''.
>
> The results are consistent with previous experiments.
> QSD-PBT CE is trained with the BD fire rate and the diversity of DvD-PPO is defined by action pairs. Therefore, both of them favor certain random BDs and perform slightly better than PPO and PBT.
> Comparing QSD-PBT and QSD-PBT CE, we may conclude that our algorithm is capable of "what you will get is what you have defined'' since it allows for the direct computation of the diversity gradient.
>
> ***
>
> > **[Q5]** Code is not included in the supplementary material.
>
> **[A5]** The code cleaning is ongoing, and we will open source once the paper is accepted.
>
> ***

---

> ### Author Response · Authors · 2022-11-17
> **Response to Reviewer 7pVo (Part 1)**
>
> ***
>
> > **[Q1]** The setting considered in the paper requires multiple design choices, which brings the practicality of the method into question. Apart from designing rewards, one needs to design a set of diversity design functions that depend on trajectories (which include a large number of states and actions) and may be difficult to elicit desired behavior.
>
> **[A1]** We would like to note that the definition of BD in this work is more general-purposed, because of its depedence on the whole trajectory. Previous works that define BD on states or actions can be regarded as special cases of our trajectory BD.
>
> In practice, users can flexibly define BDs according to their diversity demands, either on the whole trajectory or only certain states/actions. For example, users can define the BD to be the terminal position of a 2D-maze navigation problem, if the user is intesested in the diversity of the distribution of terminal state positions.  Note that this BD is allowed in our definition: ${\rm BD}(\tau) = (x_T, y_T)$.
>
> We agree with the reviewer that the design of BDs may be difficult to elicit desired behavior. But compared with previous works that use task-agnostic (implicit) BDs or task-specific (explicit) state/action BDs, our task-specific trajectory BDs may be a better or more flexible way (because previous BDs are all special cases of our BDs) to elicit user-desired behaviors.
>
> We have made it more clear in the revision, Section 2.
>
> ***
>
> > **[Q2]** Another design element is the diversity measure function (an example is given which is the determinant of a kernel matrix.)
>
> **[A2]** We do not intend to design the diversity measure function. It is applied only to measure the diversity of the population once the BDs are specified. In other words,  we need a scalar diversity objective function to be optimized.
>
> Besides, there are not many diversity measure functions (the two most common ones: MSE and DPP) studied in previous related works. Although the QSD-PBT algorithm allows the user to specify a new measure function, the evaluation in this paper focuses only on the MSE and DPP measure functions.
>
> We have made it more clear in the revision, Section 2.
>
> ***
>
> > **[Q3]** In Section 3.2, the authors mention a relationship between large $\lambda$, which determines the strength of diversity, and optimistic initial values used for exploration. This connection is unclear and the two seem rather distinct from my perspective.
>
> **[A3]** The optimistic initial value here means setting a large initial Q-value to encourage the agent to explore each action. Here in our QSD-PBT method, a large initial $\lambda_0$ for diversity loss is set to encourage the population to focus on the exploration of diverse policies at the beginning of the training.
>
> Although the thoughts behind these two techniques are similar (encourage exploration) to some extent, the implementations are different. We agree with the reviewer's concern and have deleted this connection in the revision to avoid confusion.
>
> ***

---

### Official Review · Reviewer_rdoE · 2022-10-24

**Confidence:** 2
**Correctness:** 4
**Technical Novelty And Significance:** 3
**Empirical Novelty And Significance:** 3
**Recommendation:** 6

**Clarity, Quality, Novelty And Reproducibility:**

As discussed above, I found the paper to be overall quite clear and well written. The quality of the theoretical and experimental sections seems on par with ICLR expectations. Finally, the paper addresses an interesting and - to my knowledge - new sub-problem in RL. Although the QSD problem could benefit from a stronger motivation, the overall contribution is sufficiently novel.

Based on my familiarity with the relevant literature, however, I am not certain of the novelty of the proposed algorithm.

**Strength And Weaknesses:**

# Strengths
- Clarity: this paper is very clearly written; the overall phrasing of the problem, literature review, and description of the methodology are all very clear.
- The experiments verify the usefulness of the proposed approach, and compare to a wide range of baselines.
- The main theoretical results on estimating the diversity gradient may be of independent interest.
- The details in the final QSD-PBT algorithm are investigated with a corresponding ablation study which confirms the discussion in the main paper.

# Weaknesses
- The one section I found could benefit from some clarification is $\S$ 3.3; I found the description of the mechanism by which the different $\lambda_i$ are obtained somewhat unclear. Am I correct in understanding that for each stratification level, all $\lambda_0$ are equal (set to a high value which presumably is not particularly influential)? How does this tie into the reference to $\lambda_{\infty, 1}$ and $\lambda_{\infty, 2}$?
- One potential weakness is that all evaluations of QSD-PBT use the introduced QSD-score metric. I'd be interested in whether the authors considered other ways of combining the stratified diversity terms of (1) (for example, one could consider the worst diversity score across all intervals, although the resulting optimization difficulties may be sufficient to discard this out of hand).

# Questions
- Maybe I missed this result: do you report the highest quality discovered by each algorithm (regardless of diversity)? On a related note, how are the quality intervals defined for Figure 1: are they predetermined by the range of possible rewards, or are they chosen based on the highest reward over all approaches $R_{max}$?
- Figure 1 shows that there is a trade-off between quality and diversity; however, this trade-off disappears in most of Figure 3. Do you know what this could be due to?

**Summary Of The Paper:**

This paper tackles the question of policy diversity in RL, with the specific goal of optimizing the diversity of obtained policies _within_ different quality levels (as one would, for example, want different AI difficulty levels within video games).

To do so, the authors first propose a framework that describes this specific task, which they refer to as QSD (quality-similar diversity), with an associated QSD score. This score is meant to apply to diversity metrics built upon user-specified behavior descriptors (BDs), as opposed to diversity metrics that are learned as part of the optimization process.

Second, the authors show that one can obtain unbiased estimates of the gradient of such a user-specified BD with respect to the policy (and thus, an estimate of the diversity metric's gradient).

Finally, the authors use population-based algorithms to optimize the QSD score, showing across a variety of continuous and discrete tasks that their proposed algorithm (QSD-PBT) outperforms a variety of RL algorithms with different approaches to diversity optimization.

**Summary Of The Review:**

This paper addresses the problem of learning diverse policies of differing quality. This paper is well-written, with theoretical results that may be of independent interest, and a novel population-based algorithm that is experimentally verified across a variety of tasks. Although I have a few remaining questions about the experimental evaluation, this is overall an interesting work.

---

> ### Author Response · Authors · 2022-11-17
> **Response to Reviewer rdoE (Part 2)**
>
> ***
>
> > **[Q4]** Figure 1 shows that there is a trade-off between quality and diversity; however, this trade-off disappears in most of Figure 3. Do you know what this could be due to?
>
> **[A4]** It is an important observation, and we have added the discussion in the revision, Appendix B.6.
>
> Generally speaking, the quality and the diversity are a trade-off, and higher quality usually (but not necessarily) results in lower diversity. As environments are varied, we need to analyze the trade-off case-by-case.
> As for the difference between Figure 1 and Figure 3, this may be due to the following two reasons:
>
> 1. The definition of BDs. For MuJoCo tasks in Figure 1, we define most of the BDs on joint torques, which are highly correlated to the pose control of the robots and the resulting score. Behaviors with higher quality are more restricted than those with lower quality. While for Atari games in Figure 3, we define relatively coarse-grained BDs that have lower correlations with the quality, e.g., the left or right move preference in Figure 4 does not influence the score because of the symmetry in the game map.
>
> 2. The exploration space and complexity of the environment. Compared with MuJoCo tasks, Atari games are more complex and have a larger exploration space. Taking the game RiverRaid for example, at a lower quality level (<3000), the initial river course is narrow, and the agent can only go straight. When it reaches certain scores, the agent will enter into new scenes that have a much broader course. Later scenes (meaning higher quality) provide larger diversity space, resulting in the improvement of diversity along with quality.
>
> ***

---

> > ### Comment · Reviewer_rdoE · 2022-12-06
> > **Rebuttal acknowledgement**
> >
> > I thank the authors for their detailed reply; as it stands, I am maintaining my score and support acceptance.

---

> ### Author Response · Authors · 2022-11-17
> **Response to Reviewer rdoE (Part 1)**
>
> ***
>
> > **[Q1]** The description of the mechanism by which the different $\lambda_i$ are obtained somewhat unclear. For each stratification level, all $\lambda_0$ are equal (set to a high value which presumably is not particularly influential)? How does this tie into the reference to $\lambda_{\infty, 1}$  and $\lambda_{\infty, 2}$ ?
>
> **[A1]** Yes, all $\lambda_0$ are equal for each stratification level, if we plan to optimize the diveristy at each quality level independently from zero. In reality, we optimize the diversity at each quality level sequentially in a single run, where $\lambda& is intialized to a large value and gradually decayed to 0 (this corresponds to the highest quality level).
>
> We have reorganized sections 3.2 and 3.3, deriving the adaptive diversity loss step by step and making the notations and descriptions as clear as possible.
> In summary, we describe our algorithm step by step as follows:
>
> 1. Optimizing the diversity at one quality level, in which we train the loss function with the coefficient $\lambda$ starting from $\lambda_0$ and gradually decayed to a target value $\lambda_{\infty}$.
>
> 2. Solving the QSD problem requires maximizing diversity at multiple quality levels. This means we should train the loss function with a set of target values $\lambda_{\infty, 1}$, $\lambda_{\infty, 2}$, ...
>
> 3. Given two targets $\lambda_{\infty,1} > \lambda_{\infty,2}$,
> after we finished the optimization with $\lambda_{\infty,1}$, it may be more computatinally efficient that we continue the optimization with $\lambda_{\infty,1}$ as the intial value for optimizing the diversity at the quality level targeted by $\lambda_{\infty,2}$  than re-initializing the training with initial $\lambda_0$.
>
> 4. Finally, if $\lambda$ is decayed slowly enough from a large value $\lambda_0$ to $0$, then the training process can have enough time to fully opitmize the diversity at each target values $\lambda_{\infty, 1}$, $\lambda_{\infty, 2}$, ... in a single run. In practice, we found this much more efficient than separate training with multiple target values.
>
> ***
>
> > **[Q2]** All evaluations of QSD-PBT use the introduced QSD-score metric. Whether the authors considered other ways of combining the stratified diversity terms of (1), for example, one could consider the worst diversity score across all intervals, although the resulting optimization difficulties may be sufficient to discard this out of hand.
>
> **[A2]** Thanks for your suggestion. We would like to note that we have evaluated QSD-PBT using two types of QSD-score metrics: the summation of all quality levels and the summation of medium-to-high quality levels. Please refer to the "score > 60%" rows in Table 1 and Table 2.
>
> We agree with the reviewer that if we consider the worst diversity score across all intervals (a more difficult max-min objective), the optimization algorithm should be adapted accordingly.
> However, since higher quality may reduce the space of diversity, as shown in Figure 1, optimizing the worst diversity score across all intervals would be limited to optimizing the diversity at certain levels, which is inconsistent with the goal in this paper that we prefer diversity policies from low to high qualities.
>
> Therefore, we believe that the summation of all intervals is a relatively general and practical metric, and the QSD-PBT algorithm is proposed accordingly.
>
> ***
>
> > **[Q3]** Do you report the highest quality discovered by each algorithm (regardless of diversity)? How are the quality intervals defined for Figure 1: are they predetermined by the range of possible rewards, or are they chosen based on the highest reward over all approaches?
>
> **[A3]** We only evaluate and report the diversity from low to the optimal quality $R_{max}$.
> For each algorithm, the training will be stopped when the average quality of the population reaches  $R_{max}$, then we can compare the convergence rate by calculating the time overhead.
>
> The optimal quality $R_{max}$ is predetermined by training a quality-driven (regardless of diversity) method, i.e., TD3 for MuJoCo and PPO for Atari. Afterward, $R_{max}$ is partitioned into $M=10$ disjoint intervals, and these intervals are applied to the evaluation of all approaches in Figure 1 and Figure 3.
>
> We have made it clear in the revision, Section 4.
>
> ***

---

### Author Response · Authors · 2022-11-17
**Summary of the response**

Summary of the response:

We thank the reviewers for their insightful and thorough feedback.
We are encouraged that reviewers found our problem setting interesting and novel (R1, R2, R3), our theoretical results of independent interest (R1, R3), our experimental evaluations solid (R1, R2), and our paper clearly and well presented (R1, R2, R3).

There are several primary concerns from the reviewers. We have addressed them question-by-question and made changes to the paper accordingly. The main revisions of the manuscript are listed as follows:

1. Section 2 has been rewritten. For the behavior descriptor, we give more examples based on trajectories or states/actions. For the diversity measure function, we clarify its goal and design principles. Two commonly-used diversity measure functions are suggested. (R1, R2, R3)

2. Section 3.2 has been rewritten. We make the notation and the details of the training pipeline more clear. (R1)

3. Appendix B.2.1 and https://ibb.co/MnJK3pg: additional experiments on how $\lambda$ is initialized ($\lambda_0$) and decayed ($t_0$) will influence the convergence and the performance of QSD-PBT algorithm. (R3)

4. Appendix B.2.2 and https://ibb.co/9t2ZQ8f: additional experiments on how the choice of diversity measure functions (MSE or DPP) will influence the convergence and the performance of QSD-PBT algorithm. (R3)

5. Appendix B.3 and https://ibb.co/D1BKN73, additional experiments on random BDs. (R2)

If there is still anything unclear or unaddressed, please let us know. Many thanks again for the invaluable hard work of all reviewers to improve our work.

================================================================================================

**We find that the video links for Atari and MuJoCo in the paper are outdated. Since we can not revise the paper now, we update video links as follows:**

Atari: https://files.catbox.moe/63lq0k.mov

MuJoCo: https://files.catbox.moe/3hbzw9.mp4

---

### Decision · Program_Chairs · 2023-01-20

**Decision:**

Accept: poster

**Justification For Why Not Higher Score:**

This is a solid accept for a paper with a relatively narrower audience. An increase of the score would be on the basis of an argument for more general interest in the theoretical or empirical results, but I don't see a strong case for this.

**Justification For Why Not Lower Score:**

The paper makes a novel contribution in its formulation of quality-similar diversity. It is well executed and systematic.

**Metareview: Summary, Strengths And Weaknesses:**

The paper formulates a problem that the authors term Quality-Similar Diversity, which focuses on achieving diversity among similar-quality policies. The authors propose a population based reinforcement learning approach to address this problem, and evaluate their approach on MuJoCo and Atari domains.

Reviewers were generally positive about the paper, noting the interesting and novel problem formulation with potential for wider impact, the systematic empirical validation and strong empirical results. The theoretical insights that guided the development of the approach were found valuable, as was the thorough overview and categorization of existing diversity measures. The paper was generally found clear and readable.

Initial reviews noted a number of areas for improvement. In particular, specific sections of the paper could benefit from increased clarity (subsections of 3, 4). Reviewers asked questions about the practicality of the method (because multiple design choices need to be made), about the evaluation setup (in terms of the newly introduced QSD metric), and about sensitivity analysis and computational requirements). The authors provided detailed answers during the rebuttal phase. They updated the manuscript to increase clarity, and added additional experiments). Reviewer questions were well addressed, and the consensus is to recommend accepting the paper.

**Note From Pc:**

if the above contains the word "oral" or "spotlight" please see: "oral" presentation means -> notable-top-5% and "spotlight" means -> notable-top-25%. As stated in our emails, we are disassociating presentation type from AC recommendations